# Lysosome as a Chemical Reactor

**DOI:** 10.3390/ijms262311581

**Published:** 2025-11-29

**Authors:** Mahendiran Dharmasivam, Busra Kaya

**Affiliations:** Institute for Biomedicine and Glycomics, Griffith University, Gold Coast, QLD 4215, Australia

**Keywords:** lysosome, acidic organelles, metal-mediated reactive oxygen species, fenton reaction, lysosomotropic design, drug resistance, drug–lysosome interactions, therapeutic design principles

## Abstract

The lysosome is no longer viewed as a simple degradative “trash can” of the cell. The lysosome is not only degradative; its acidic, redox-active lumen also serves as a chemical “microreactor” that can modulate anticancer drug disposition and activation. This review examines how the distinctive chemical features of the lysosome, including its acidic pH (~4.5–5), strong redox gradients, limited thiol-reducing capacity, generation of reactive oxygen (ROS), diverse acid hydrolases, and reservoirs of metal ions, converge to influence the fate and activity of anticancer drugs. The acidic lumen promotes sequestration of weak-base drugs, which can reduce efficacy by trapping agents within a protective “safe house,” yet can also be harnessed for pH-responsive drug release. Lysosomal redox chemistry, driven by intralysosomal iron and copper, catalyzes Fenton-type ROS generation that contributes to oxidative damage and ferroptosis. The lysosome’s broad enzyme repertoire enables selective prodrug activation, such as through protease-cleavable linkers in antibody–drug conjugates, while its membrane transporters, particularly P-glycoprotein (Pgp), can sequester chemotherapies and promote multidrug resistance. Emerging therapeutic strategies exploit these processes by designing lysosomotropic drug conjugates, pH- and redox-sensitive delivery systems, and combinations that trigger lysosomal membrane permeabilization (LMP) to release trapped drugs. Acridine–thiosemicarbazone hybrids exemplify this approach by combining lysosomal accumulation with metal-based redox activity to overcome Pgp-mediated resistance. Advances in chemical biology, including fluorescent probes for pH, redox state, metals, and enzymes, are providing new insights into lysosomal function. Reframing the lysosome as a chemical reactor rather than a passive recycling compartment opens new opportunities to manipulate subcellular pharmacokinetics, improve drug targeting, and overcome therapeutic resistance in cancer. Overall, this review translates the chemical principles of the lysosome into design rules for next-generation, more selective anticancer strategies.

## 1. Introduction

Lysosomes, first described by De Duve in the 1950s, have long been recognized as acidic organelles enriched with hydrolases responsible for macromolecular degradation [1,2,3,4]. In cancer biology, they are traditionally highlighted for their roles in autophagy, metabolism, and cell death pathways, such as cathepsin-mediated apoptosis following lysosomal membrane permeabilization (LMP) [1,5,6,7]. However, beyond these biological functions, a paradigm shift is emerging that redefines lysosomes as chemical reactors with distinct physicochemical characteristics that can be harnessed in drug discovery. This concept is particularly relevant in cancer, where lysosomal properties can critically determine the fate and efficacy of chemotherapeutic agents [8,9,10,11].

In tumor cells, lysosomes are often more numerous and enlarged, reflecting enhanced autophagy and lysosomal biogenesis driven by cellular stress and oncogenic signaling [12,13,14,15]. These altered lysosomes present both challenges and opportunities: they can sequester and inactivate anticancer drugs, reducing therapeutic efficacy, yet also provide a unique chemical environment that can be harnessed for selective drug activation [16,17,18]. Notably, cancer-associated changes such as the upregulation of lysosomal membrane proteins, disrupted pH gradients, and dysregulated metal metabolism distinguish tumor lysosomes from their normal counterparts [1,17,19,20,21,22,23]. These differences position the tumor lysosome itself as a promising therapeutic target.

This review advances a perspective centered on the lysosome’s chemical environment and its influence on cancer drug discovery and therapy (Figure 1). We first examine the interdependent physicochemical parameters of the lysosomal lumen, including acidity, redox balance, thiol pools, enzymatic content, and metal ions, and explain their hierarchy. The acidic pH (~4.5–5.0) provides the foundation by dictating enzyme activity profiles, protonation states, and metal ion solubility. Within this environment, the redox potential, maintained by the glutathione (GSH) GSH/GSSG couple and enzymes such as γ-interferon-inducible lysosomal thiol reductase (GILT), controls thiol reactivity and reactive oxygen species chemistry, which together regulate oxidative and reductive processes.

Metal ions such as iron (Fe) and copper (Cu) connect these parameters to catalytic transformations and chemical speciation, while the repertoire of acid hydrolases executes pH-dependent reactions whose rates are further influenced by the local redox and metal status [24]. These factors act as a coordinated network rather than independent features, creating a chemically distinctive microreactor that shapes drug fate. Subsequent sections analyze how this microenvironment governs drug behavior, from unintended trapping and degradation to deliberate activation and release. Later sections focus on therapeutic strategies that intentionally harness lysosomal chemistry, including exploiting pH gradients for drug delivery, designing lysosomotropic agents, and developing metal-based redox approaches that convert lysosomes into Trojan-horse toxin generators. Additional strategies aim to overcome the lysosomal barrier in multidrug-resistant cancer.

Enabling technologies and chemical biology tools that illuminate these processes are highlighted, along with key examples such as acridine–thiosemicarbazone hybrids [8] and P-glycoprotein (Pgp) repurposing, which demonstrate the translational potential of treating the lysosome as a chemical reactor [8]. The review concludes by outlining future directions and emphasizing how a deeper chemical understanding of lysosomes could guide the development of next-generation, more selective and effective anticancer therapies.

## 2. The Lysosomal Chemical Microenvironment in Cancer

Lysosomes establish an interior milieu that differs profoundly from the cytosol [25,26]. These distinctions are both quantitative, encompassing extreme acidity, distinct redox potential, and altered ion concentrations, and qualitative, involving the presence of specialized enzymes and transition metals [8,9,27]. Together, these factors enable chemical reactions that would be inefficient or impossible elsewhere in the cell. The following sections highlight the major facets of this microenvironment, with particular attention to how cancer cell lysosomes amplify or exploit these unique chemical features.

Overall, the lysosome functions as an integrated chemical reactor rather than a collection of isolated conditions. Acidic pH generated by the V-ATPase sets the protonation state of weak bases, controls hydrolase activity, and influences the solubility and speciation of Fe and Cu (Figure 1). Within this acidic lumen, redox-active metals, low-molecular-weight thiols, and hydrogen peroxide interact to sustain Fenton and Fenton-like reactions that generate ROS. At the same time, pH-sensitive linkers and enzyme substrates in antibody–drug conjugates and other carriers are cleaved during endosomal–lysosomal trafficking, releasing active payloads. As summarized in Figure 1, these interconnected processes determine whether drugs are sequestered and functionally inactivated, locally activated within lysosomes, or cooperate with ROS to damage membranes and trigger cell death.

Importantly, lysosomal properties differ substantially between normal and cancer cells, and these variations strongly influence drug behavior. Many tumors display deeper luminal acidification, increased V-ATPase activity, enlarged lysosomal volume, and elevated expression of cathepsins and other hydrolases, which together enhance weak-base trapping and accelerate degradation or activation of pH- or enzyme-sensitive linkers. Cancer cells also accumulate larger pools of labile Fe and Cu due to heightened autophagy and altered metal metabolism, predisposing their lysosomes to Fenton-type ROS production and redox cycling. In contrast, normal cells maintain tighter control over luminal pH, metal availability, and hydrolase expression, creating a less reactive environment. These tumor-specific features establish a chemically more potent and drug-responsive lysosomal compartment that shapes differential sequestration, redox sensitivity, and susceptibility to lysosome-dependent cell death.

### 2.1. Acidic pH and Protonation Dynamics

The lysosomal lumen maintains a pH of approximately 4.5–5.0, in sharp contrast to the near-neutral cytosolic pH of ~7.2 [28,29,30,31]. This pronounced proton gradient, established by the V-ATPase proton pump, is essential for hydrolase activation and substrate turnover [32,33]. Chemically, the low pH profoundly influences drug protonation and solubility [34]. Weak-base molecules with pKa values in the range of 6–9 readily gain protons in the lysosome, becoming charged and membrane-impermeable (Figure 1) [35,36,37]. These species accumulate to high concentrations inside lysosomes through a process known as acidic trapping or weak-base sequestration [12].

For example, the tyrosine kinase inhibitor Sunitinib (Figure 2A), a hydrophobic weak base, concentrates in cancer cell lysosomes, correlating with intrinsic resistance [12,38,39]. Similarly, Doxorubicin (Figure 2B) and Mitoxantrone (Figure 2C) redistribute into acidic vesicles, limiting their nuclear delivery (Figure 3) [40,41]. Lysosomal trapping can dramatically reduce the cytosolic concentration of some drugs, effectively partitioning a substantial fraction of the total dose into the acidic compartment (Figure 3) [31,35,42,43]. This phenomenon has been shown to promote adaptive cellular responses that further enhance lysosomal capacity and drug sequestration [18,44]. Zhitomirsky and Assaraf demonstrated that nanomolar doses of weakly basic drugs stimulate cancer cells to increase lysosome numbers through Transcription Factor EB (TFEB)-mediated biogenesis, effectively expanding the lysosomal “sink” and compounding multidrug resistance (MDR) over time [12,45,46].

From a chemical perspective, acidic pH also alters the stability of pH-sensitive bonds [34]. Many hydrolytic reactions are proton-catalyzed, so acid-labile functional groups such as imines, hydrazones, and acetals cleave more rapidly in lysosomes than at neutral pH [47]. Medicinal chemists have exploited this principle to design pH-cleavable linkers that remain stable in circulation but release their payloads within lysosomes [34,48,49]. For activation, Doxorubicin-bearing hydrazone linkers provide a representative case, as they hydrolyze more rapidly at lysosomal pH to liberate the active drug. For inactivation, weakly basic agents such as Sunitinib, Doxorubicin, and Mitoxantrone become protonated and sequestered within lysosomes, reducing their cytosolic or nuclear availability (Figure 2 and Figure 3). Conversely, strongly acidic conditions can inactivate drugs that are unstable to protonation, emphasizing the need to design agents that either tolerate low pH or avoid prolonged lysosomal residence [48,50].

The protonation state within lysosomes also modulates metal chelation and redox reactions, as proton availability can influence Fenton-type chemistry (Figure 1) [51]. In cancer cells, lysosomal pH can vary slightly. Some studies report modest alkalinization (pH ~5) under stress or high metabolic demand, whereas others indicate that aggressive tumors maintain strong acidification [52]. Notably, a one-unit change in pH corresponds to a tenfold difference in proton concentration, meaning even small fluctuations can significantly affect drug trapping and enzymatic activity. Overall, acidic pH defines the lysosome’s identity as a chemical microreactor, driving protonation-dependent drug sequestration and accelerating acid-catalyzed transformations central to cancer pharmacology [53].

### 2.2. Redox Conditions and Reactive Species

The lysosomal lumen maintains a unique redox balance [54]: it is comparatively oxidizing yet contains selective reductants and lacks major antioxidant defenses. Several factors contribute to this environment. Lysosomes accumulate redox-active metals such as iron and copper through the degradation of metalloproteins and transferrin uptake, allowing continuous cycling between oxidation states (Table 1) [55,56,57]. The catabolism of macromolecules also generates hydrogen peroxide (H_2_O_2_) and superoxide, partly via enzymes such as monoamine oxidase in mitochondrial membrane fragments and through peroxidase-like activity during heme degradation [58]. Critically, lysosomes are deficient in antioxidant enzymes such as catalase and glutathione peroxidase [1,59,60]. Consequently, any H_2_O_2_ that enters or forms within the lumen is not efficiently neutralized, unlike in peroxisomes or the cytosol, where catalase and peroxidases tightly regulate peroxide levels [61].

This combination makes lysosomes hotspots for Fenton chemistry [69,70,71]. The Fenton reaction (Fe^2+^ + H_2_O_2_ → Fe^3+^ + •OH + OH^−^) produces highly reactive hydroxyl radicals (•OH) and proceeds most rapidly under acidic conditions (Table 1) [72]. Lysosomes provide both the required acidity and a supply of Fe^2+^ and H_2_O_2_. Despite the oxidizing environment, lysosomes also contain low-molecular-weight reductants such as cysteine, glutathione (GSH), and ascorbate that regenerate Fe^2+^ from Fe^3+^, thereby sustaining redox cycling (Figure 4) [1,73,74,75]. In this way, the lysosome functions as a redox reactor generating reactive oxygen species (ROS), including hydroxyl radicals and copper-dependent radicals through Fenton-like Haber–Weiss reactions (Figure 4) [76,77,78]. These ROS oxidize lipids, proteins, and nucleic acids in proximity to the lysosomal membrane [79,80,81].

Lipid peroxidation of lysosomal membranes is now recognized as a critical initiating event in ferroptosis, an iron-dependent cell death pathway [82,83]. Recent studies have shown that the ferroptosis inducer RAS-selective lethal 3 (RSL3; Figure 5) promotes lipid peroxidation within lysosomes by driving iron-dependent oxidative reactions. In contrast, lipophilic antioxidants such as Coenzyme Q10 (CoQ10) [84,85] or vitamin E (α-tocopherol) [86,87], which act as membrane-embedded radical-trapping antioxidants, can suppress ferroptosis by neutralizing reactive oxygen species and preventing intralysosomal iron activation [82,88]. Synthetic radical-trapping antioxidants such as ferrostatin-1 and liproxstatin-1 also inhibit ferroptosis by blocking lipid peroxidation. Similarly, a synthetic molecule named fentomycin-1 was designed to activate lysosomal iron directly, triggering extensive phospholipid oxidation and selective death of iron-rich cancer cells [82]. These findings highlight that the site of redox activity is crucial: the lysosome is not merely a bystander in oxidative stress but can serve as a deliberate target for pro-oxidant cancer therapies.

Lysosomal redox conditions are also influenced by the availability of thiols and disulfides [89]. Whereas the cytosol is strongly reducing, containing millimolar concentrations of GSH and abundant NADPH, the lysosome maintains a much lower thiol concentration [90]. Reduced GSH does not freely cross the lysosomal membrane and becomes largely protonated and membrane-impermeable under acidic conditions, resulting in free GSH levels several orders of magnitude lower than those in the cytosol, typically in the low micromolar range. Cysteine enters the lysosome through cysteine/cystine transporters such as cystinosin, but it can rapidly oxidize to cystine or be consumed in reactions at acidic pH (Figure 4) [91,92].

Notably, cells express a specialized enzyme, γ-interferon-inducible lysosomal thiol reductase (GILT), which is the only known enzyme that catalyzes disulfide bond reduction in the endocytic pathway [93]. GILT operates optimally at pH ~5 and uses a Cys–His–Asp catalytic triad to cleave disulfide bonds in internalized proteins, promoting their unfolding and degradation (Table 1). It also plays a crucial role in reducing disulfide-linked antigens during major histocompatibility complex (MHC) class II processing, thereby linking lysosomal redox chemistry to immune signaling.

This system confers a finely tuned reductive capacity, sufficient for specific disulfide cleavage when catalyzed by GILT, but far weaker than the broadly reducing conditions of the cytosol [94]. In GILT-deficient cells, the lysosomal environment becomes more oxidizing, with elevated superoxide levels and oxidized glutathione, confirming GILT’s role in maintaining redox balance within the lumen [93]. The lysosome thus sustains an oxidative–reductive equilibrium: oxidative enough to support ROS chemistry and preserve disulfide stability, yet modestly reductive to enable selective bond cleavage.

This controlled redox capacity directly affects drug stability and activation. Redox-sensitive linkers or conjugates, such as disulfide bonds in nanoparticles or prodrugs, may undergo cleavage by GILT or by intralysosomal thiols if the potential is sufficiently reducing [49,95,96]. However, most disulfide-based release systems are designed for the cytosol’s stronger reducing environment [97], highlighting the importance of aligning chemical triggers with the correct intracellular compartment. Understanding and matching redox responsiveness to lysosomal versus cytosolic chemistry is therefore critical for designing effective redox-activated therapeutics [98,99].

### 2.3. Metal Ion Sequestration and Catalysis

Lysosomes play a central role in cellular metal ion homeostasis, sometimes referred to as “metallosomes” [100]. When cells internalize ferritin or other metalloproteins through autophagy or endocytosis, lysosomal enzymes degrade the protein shell and release ions such as iron and zinc into the lumen [55,101]. These ions form a pool of labile metals. For iron, much of this pool complexes with phosphate or precipitates as hemosiderin-like aggregates, but a fraction remains redox-active and exchangeable between Fe^2+^ and Fe^3+^ [102].

This labile iron pool is particularly relevant in cancer [103]. First, cancer cells often exhibit dysregulated iron metabolism, a phenomenon described as “iron addiction” [104]. Rapidly proliferating tumors internalize more transferrin-bound iron, leading to accumulation in lysosomes [55]. These iron-rich lysosomes can act as potential “time bombs” for Fenton chemistry [105]. Under physiological conditions, controlled iron redox cycling within lysosomes supports metabolic and signaling processes essential for tumor growth. However, during oxidative stress or partial lysosomal membrane permeabilization, Fenton-derived radicals (•OH) can escape into the cytosol, amplifying oxidative signaling, DNA damage, and inflammation. The association between lysosomal iron and cell vulnerability is exemplified in ferroptosis, where iron-dependent oxidative reactions trigger lipid peroxidation and cell death [82,106]. Importantly, chronic exposure to sub-lethal oxidative stress can paradoxically promote cancer cell adaptation by activating NRF2-driven antioxidant programs, enhancing metabolic flexibility, and supporting tumor proliferation, invasion, and therapy resistance [107,108].

Second, lysosomal iron also contributes to oxidative stress in non-cancer contexts by catalyzing the formation of lipofuscin, an oxidized protein–lipid aggregate that accumulates with age [109]. In tumors, elevated oxidative stress may produce similar damage, resulting in lipofuscin-like buildup that impairs lysosomal function and increases membrane fragility [110].

Copper is another redox-active metal enriched in lysosomes [24,55,56,111]. In Wilson’s disease, excess copper accumulates in hepatic lysosomes due to defective copper transport [112,113]. In cancer, copper is essential for angiogenesis and other growth processes, and many tumors upregulate copper uptake [114]. Copper-binding drugs and copper-based nanoparticles are often trafficked into lysosomes via endocytosis [111,115]. Within this acidic environment, copper participates in Fenton-like reactions (Cu^+^ + H_2_O_2_ → Cu^2+^ + •OH + OH^−^), with Cu^2+^ reduced back by protons or thiols (Figure 1) [116].

This chemistry has inspired chemodynamic therapy (CDT), which exploits lysosomal acidity to promote ROS formation [1]. Studies show that Cu^2+^-based nanoagents often catalyze these reactions faster under acidic conditions than traditional Fe-based systems [117]. For example, copper peroxide nanoparticles dissolve in lysosomes to co-release Cu^2+^ and H_2_O_2_, driving localized ROS generation (Figure 1) [1,118]. In one design, tuning nanoparticle surface pKa controlled lysosomal residence time, with lower-pKa surfaces favoring retention, enhanced ROS generation, and improved tumor cell killing [119]. These examples highlight how lysosomal metal chemistry can be harnessed therapeutically, as lysosomes provide both substrates (H_2_O_2_) and conditions (acidity and absence of catalase) favorable for metal-catalyzed radical formation.

Other metals, including zinc and manganese, also accumulate in lysosomes [120,121,122]. Although Zn^2+^ is not redox-active, it can modulate enzyme activity and signaling [123]. Zinc frequently complexes with degraded macromolecules, and its dysregulation in lysosomes has been linked to neurodegenerative diseases, although its role in cancer remains less defined [124]. Calcium is another important ion stored in acidic organelles [125]. Lysosomal Ca^2+^ release can signal exocytosis or membrane repair [126]. Certain therapies, such as ionophores or specific nanoparticles, can trigger lysosomal calcium release, potentially inducing cell death or stress responses [127].

From a drug design perspective, the lysosomal metal pool offers new opportunities for metal-based therapeutics [8,9,24,128]. Some experimental thiosemicarbazones form complexes with Fe^3+^ or Cu^2+^ upon entering cells [128,129,130]. Within lysosomes, these complexes can undergo redox cycling to generate ROS, deliberately inducing lysosomal damage [9]. Gold-based compounds also display lysosomal localization [131,132]. While gold(III) complexes and gold(I)–phosphine complexes are generally known to react with thiols in cytosolic or mitochondrial enzymes, their cationic and lipophilic properties can drive accumulation in lysosomes, where they induce macromolecular crosslinking and inhibit protease activity [133,134].

Gold(III)–thiosemicarbazone complexes are particularly interesting, as gold(III) is a soft Lewis acid that can be reduced to gold(I) or elemental gold by thiols, a process favored under acidic conditions [135,136]. Such redox transformations may enable selective activation or precipitation of gold complexes within lysosomes, disrupting their membranes and promoting cancer cell death.

Lysosomes act as key sites for metal ion sequestration and catalysis, coordinating redox-active and redox-inert metals alike. This intra-organellar “metal chemistry” can be detrimental by promoting oxidative damage or can be therapeutically exploited in strategies such as chemodynamic therapy or metal-based prodrugs [137]. Understanding the lysosomal metallome—which metal species are present and in what redox states—is crucial for the rational design of next-generation anticancer agents that operate within this chemically distinct compartment.

### 2.4. Enzymatic Hydrolysis: Acid Hydrolases as Catalysts

The lysosome houses a diverse repertoire of acid hydrolases, comprising more than 60 enzymes that include proteases such as cathepsins B, D, and L, glycosidases, lipases, phosphatases, sulfatases, and nucleases [138]. These enzymes are optimized to function at acidic pH, transforming the lysosome into a catalytic hub capable of degrading nearly every type of biological macromolecule [1]. In cancer, lysosomal hydrolases often exhibit increased expression or secretion [139]. Cathepsins B and L, for instance, are frequently upregulated in invasive tumors and secreted extracellularly to facilitate matrix degradation [140]. Within lysosomes, these enzymes also influence drug fate by activating, deactivating, or degrading therapeutic molecules.

Antibody–drug conjugates (ADCs) and nanoparticle-based carriers frequently rely on lysosomal enzymes to release active drugs (Table 1) [141,142,143]. Peptide linkers can be cleaved by cathepsin B, while ester linkages are hydrolyzed by acid lipases [49,66,144]. However, this enzymatic activity can also be detrimental if a drug or carrier is prematurely degraded. Liposomal or polymeric micelle formulations, for example, can be engulfed by lysosomes and enzymatically dismantled, resulting in unintended drug release within the lysosomal lumen rather than at the intended intracellular site [145,146].

Conversely, prodrugs designed to be enzyme-cleavable intentionally leverage lysosomal enzymes for activation. A prominent example is the cathepsin B-cleavable dipeptide linker valine–citrulline (Val–Cit), used in several FDA-approved ADCs such as brentuximab vedotin (Figure 6) [141,147,148]. This linker remain stable in circulation but are cleaved within lysosomes at the Cit–PABC junction, releasing the cytotoxic payload monomethyl auristatin E [147,149]. This ensures that drug activation occurs only after the antibody–drug complex has reached the lysosome of the target cell, thereby minimizing off-target toxicity [147,150].

Similar strategies extend to small-molecule prodrugs and polymer–drug conjugates, including those using β-glucuronidase-sensitive linkers that exploit lysosomal β-glucuronidase activity, often elevated in necrotic tumor regions [151]. Matrix metalloproteinase-sensitive peptides have also been employed, although these enzymes typically act extracellularly in the tumor microenvironment rather than within lysosomes [152].

Importantly, lysosomal enzyme expression and activity can vary significantly across tumor types [153]. Some cancers display altered isoforms or expression levels of hydrolases, influencing the efficiency of drug activation. For example, while many ADCs rely on cathepsin B as the primary activating enzyme, studies have shown that other lysosomal cathepsins can compensate when cathepsin B is absent, reflecting the functional redundancy of these proteases [154]. This redundancy is advantageous for robust drug activation but can also complicate specificity. Moreover, if a tumor downregulates a target enzyme or if lysosomal leakage occurs, extracellular drug release may lead to off-target effects.

From a chemical perspective, lysosomal enzymes function as a broad suite of acid-tolerant catalysts capable of executing parallel hydrolysis reactions [155]. Their substrate promiscuity, particularly among cathepsins B and L, can be advantageous for activating prodrugs but poses challenges for drug stability [66]. Any compound containing cleavable bonds, including peptide, glycosidic, ester, or acetal linkages, may be susceptible to enzymatic degradation [156]. Consequently, medicinal chemists often design small-molecule drugs to resist lysosomal hydrolysis unless activation is intentionally desired [157].

The lysosomal enzyme network underpins the organelle’s identity as a multifunctional catalytic reactor [33]. It confers versatility in processing diverse chemical substrates, a feature increasingly exploited in modern drug design to achieve controlled activation and enhanced therapeutic precision.

## 3. Lysosomotropic Drug Accumulation and the “Safe House” Effect

Many small-molecule anticancer agents are cationic, amphiphilic weak bases at physiological pH [8,12,158,159,160], Representative examples include Doxorubicin (Figure 2B) [12,161,162,163], which contains an amino sugar; tyrosine kinase inhibitors such as Imatinib (Figure 7A) [164,165,166] and Sunitinib (Figure 2A) [166,167,168]; and Chloroquine (Figure 7B) [169,170,171], a repurposed antimalarial now used as an autophagy inhibitor in oncology trials. These compounds can diffuse across membranes in their neutral form but become protonated within acidic organelles, rendering them membrane-impermeable.

As a result, they accumulate preferentially in lysosomes and late endosomes, often reaching concentrations several hundred times higher than those in the cytosol or extracellular medium. This accumulation mechanism, known as acidic trapping or lysosomotropism, is vividly illustrated by fluorescent weak bases such as Acridine Orange or Lysotracker dyes, which selectively label lysosomes [172]. In cancer cells, doxorubicin’s red fluorescence frequently localizes to punctate lysosomal structures in resistant cells, rather than the diffuse nuclear distribution observed in sensitive cells [9].

This phenomenon underlies the “lysosomal safe house” model of drug resistance, in which cancer cells exploit lysosomes as protective reservoirs [9,173]. Drugs sequestered in lysosomes are physically isolated from their primary intracellular targets, such as topoisomerase II, DNA, or kinase enzymes, and can be expelled from the cell through lysosomal exocytosis [12,160,174]. Chronic drug exposure often drives expansion of the lysosomal compartment through TFEB-mediated lysosomal biogenesis, further enhancing drug sequestration [12]. This represents a distinct form of intrinsic MDR that arises not from drug efflux at the plasma membrane or target mutation, but from intracellular compartmentalization. Studies have shown that exposing cancer cells to low doses of weakly basic drugs such as Mitoxantrone produces resistant clones with enlarged lysosomal vesicles densely packed with drug, demonstrating how lysosomal expansion can reinforce resistance.

A related mechanism involves Pgp, the well-known plasma membrane efflux transporter [175]. Under certain stress conditions, such as glucose deprivation or prolonged drug exposure, Pgp can localize to lysosomal membranes [173,176,177], where it actively transports substrates from the cytosol into lysosomes [9,178]. Doxorubicin (Figure 2B), a classic Pgp substrate, exhibits co-localization with lysosomal markers in Pgp-overexpressing cells, while Pgp-deficient cells show nuclear localization [179]. This intracellular trafficking adds another layer to lysosomal sequestration, as Pgp can effectively load drugs into lysosomes [9].

However, the magnitude of this effect remains debated. Some analyses suggest that the lysosomal Pgp mechanism requires unrealistically high transporter activity and drug concentrations to generate meaningful gradients [180]. Excessive accumulation may also destabilize lysosomal membranes, making the process self-limiting. Thus, while lysosomal Pgp contributes to drug sequestration, passive pH trapping remains the dominant driver of this phenomenon in most systems [180].

Recognizing lysosomal trapping as a contributor to drug resistance has inspired multiple therapeutic strategies. One approach seeks to prevent trapping by alkalinizing lysosomes, thereby promoting drug redistribution to the cytosol [181]. Agents such as chloroquine (Figure 7B), hydroxychloroquine (Figure 7C), and proton pump inhibitors have been tested clinically to raise lysosomal pH, though complete neutralization can impair essential cellular processes and lacks tumor selectivity [182,183].

Another strategy is to circumvent trapping through chemical design. Medicinal chemists can reduce a compound’s basicity by lowering its pKa or by introducing polar substituents, thereby limiting its lysosomal accumulation and preserving cytosolic exposure. Alternatively, some therapeutic designs exploit trapping intentionally, combining lysosomotropic drugs with secondary agents that destabilize lysosomes once loaded. Such combination approaches convert the lysosomal “safe house” into a cytotoxic trigger, a concept explored further in subsequent sections.

### 3.1. Lysosomal Drug Metabolism and Activation

Beyond serving as sites of drug sequestration, lysosomes can also participate in drug activation and metabolism. Many targeted and nanoparticle-based therapeutics are deliberately designed to exploit lysosomal conditions for controlled drug release.

A prime example is ADCs, which rely on lysosomal proteases to cleave linkers and liberate the cytotoxic payload [147,184,185]. If lysosomal enzyme activity is reduced or if the payload fails to separate from the antibody, the ADC remains inactive. Some resistant cancer cells have been shown to alter lysosomal enzyme expression or luminal pH, thereby preventing efficient drug release. This represents a distinct resistance mechanism in which the lysosome fails to perform the expected chemistry. Conversely, tumor cells with elevated cathepsin activity may process ADCs more efficiently, leading to greater payload release and enhanced cytotoxicity [147,154]. Consequently, patient-to-patient variability in lysosomal enzyme composition and activity could significantly influence clinical responses to lysosome-dependent therapies.

Several nanoparticle formulations are designed to respond to lysosomal cues such as acidity or enzymatic activity. pH-sensitive liposomes, often incorporating pH-responsive lipids or polymers, remain stable at physiological pH (7.4) but undergo phase transitions or membrane disruption under mildly acidic conditions (pH 5–6). This behavior promotes localized drug release in endosomes or lysosomes, enhancing cytosolic delivery or preventing premature drug loss. For example, polymeric nanocarriers incorporating hydrazone linkers can stably retain doxorubicin in circulation yet release it once the hydrazone bond hydrolyzes in acidic endo/lysosomal environments [186]. Such systems effectively use the lysosome as a chemical trigger, where hydrolysis acts as a programmed release mechanism for therapeutic activation.

Lysosomes can also contribute to drug metabolism through enzymatic degradation [157]. Lipid-like molecules and peptides are frequent substrates of lysosomal hydrolases, and PEGylated peptides or polymer conjugates may undergo proteolytic cleavage once internalized. An illustrative case is albumin-bound paclitaxel (nab-paclitaxel; Abraxane) [187]. Tumor cells internalize albumin via macropinocytosis and degrade it in lysosomes, releasing bound paclitaxel. This process was initially proposed as a tumor-targeting mechanism, as many cancers exhibit increased albumin uptake.

However, in tumors with highly active lysosomal catabolism, rapid degradation of albumin carriers may release drugs within lysosomes, potentially leading to secondary sequestration if the payload is protonatable. While paclitaxel itself is not a weak base and can readily diffuse out, other albumin-bound chemotherapeutics might be retained in the lysosomal lumen following release [188].

Lysosomes play a dual role in pharmacology [189]: they can activate drugs through enzymatic or pH-dependent mechanisms and inactivate them through degradation or sequestration. Understanding this balance is critical for the rational design of next-generation therapeutics that either harness or avoid lysosomal metabolism to optimize efficacy.

### 3.2. Lysosomal Membrane Permeabilization: A Double-Edged Sword

LMP refers to the loss of integrity of the lysosomal membrane, ranging from minor leaks to complete rupture [190,191]. Chemically, LMP can result from high intralysosomal ROS that oxidize membrane lipids, accumulation of detergent-like drugs or lipids, or exposure to membrane-destabilizing agents. In pharmacological terms, LMP represents a dramatic turning point because the sudden release of lysosomal contents, including proteases, ROS, and sequestered drugs, into the cytosol often triggers cell death through cathepsin-mediated apoptosis or necroptosis, or through widespread oxidative damage [14].

In cancer therapy, the selective induction of LMP has emerged as an attractive strategy to eliminate tumor cells that are resistant to apoptosis [192]. Conventional therapies typically rely on caspase-dependent pathways, which are frequently disrupted in cancer. In contrast, rupturing lysosomes unleashes stored hydrolases and ROS, destroying the cell through caspase-independent mechanisms [1]. The major challenge is achieving tumor selectivity by inducing LMP in malignant cells without harming normal tissue. Nanotechnology has provided promising solutions [1,193].

For instance, engineered nanoparticles can preferentially accumulate in tumor lysosomes or activate under tumor-specific conditions, such as acidic pH or elevated enzyme expression [194]. One example used magnetic nanoparticles that localized to tumor lysosomes. When exposed to an alternating magnetic field, the particles oscillated and generated localized heat and mechanical stress, rupturing lysosomal membranes from within. Normal cells, which lacked particle accumulation, were unaffected [195].

Certain small molecules can also trigger LMP. The lysosomotropic detergent Siramesine (Figure 8A), a cationic amphiphilic compound, accumulates in lysosomes and solubilizes membranes at high concentrations, killing cancer cells through lysosomal cell death [196]. Although preclinical data were promising, off-target toxicity limited its development. Arsenic trioxide induces LMP in leukemia cells, potentially by binding membrane proteins or generating ROS that damage the lysosomal membrane [197]. Natural products such as Saponins (Figure 8B) and Indole alkaloids (Figure 8C) have also been shown to permeabilize lysosomes through detergent-like or oxidative mechanisms [198].

From a chemical standpoint, a drug’s ability to cause LMP often correlates with the extent of its lysosomal accumulation and its capacity to generate stress within the lysosome, particularly ROS. Compounds capable of redox cycling, producing superoxide or peroxides in acidic conditions, are potent LMP inducers. Thiosemicarbazones complexed with copper exemplify this mechanism, as they promote intralysosomal ROS formation leading to membrane rupture [24]. This property can be harnessed therapeutically. Combining LMP-inducing agents such as di-2-pyridylketone-4,4-dimethyl-3-thiosemicarbazone (Dp44mT) or di-2-pyridylketone-4-cyclohexyl-4-methyl-3-thiosemicarbazone (DpC; Figure 9A,B) with trapped drugs like doxorubicin can release the drug back into the cytosol, restoring its cytotoxicity in multidrug-resistant cells [9,24]. This synergy highlights how understanding lysosomal drug disposition enables the rational design of combination therapies.

Not all LMP is lethal. Some cancer cells experience chronic, sub-lethal LMP, releasing small amounts of cathepsins that activate prosurvival pathways. For example, partial lysosomal permeabilization can activate the transcription factor nuclear factor erythroid-2-related factor 2 (NRF2), enhancing cellular defense against oxidative stress [199]. Cancer cells may tolerate this limited leakage as a trade-off for signaling advantages and invasive behavior. However, when lysosomal damage surpasses a critical threshold, the ensuing loss of membrane integrity becomes catastrophic and irreparable.

### 3.3. Lysosomal Inactivation of Drugs and Drug–Drug Interactions

Lysosomes can serve as intracellular compartments where drugs not only accumulate but also interact or undergo unintended chemical changes [200]. Because many therapeutics co-sequester within lysosomes, the simultaneous administration of two weak-base drugs can lead to drug–drug interactions [201]. Each compound may influence the lysosomal environment or the other’s partitioning behavior. For instance, one drug may act as a proton sponge, raising lysosomal pH and thereby reducing the accumulation of another weak base. Chloroquine (Figure 7B) and related antimalarials are classic examples; they elevate lysosomal pH and have been explored in combination with chemotherapies to enhance efficacy by promoting the release of sequestered drugs or by inhibiting autophagy [202]. In human immunodeficiency virus (HIV) therapy, lysosomal trapping explains certain interactions in which antidepressants that accumulate in lysosomes displace anti-HIV protease inhibitors, altering their effective intracellular concentrations [203].

Beyond physical sequestration, drugs can also undergo chemical modification within lysosomes. Although the lysosome lacks drug-metabolizing enzymes such as cytochrome P450s, simple chemical reactions can occur under its acidic and oxidative conditions [204]. Examples include the hydrolysis of β-lactam antibiotics by lysosomal proteases or condensation reactions involving Schiff-base-forming drugs and aldehydes derived from lipid peroxidation. While such reactions are relatively uncommon, they may be significant for specific chemotypes that are chemically labile under acidic or oxidative conditions.

Overall, the lysosomal environment can act as a site of drug processing, inactivating drugs, retaining them for prolonged periods, or promoting chemical alteration. Consequently, modern drug discovery increasingly incorporates assessment of lysosomal behavior through parameters such as the lysosomal trapping index and redox stability profiling. Understanding these factors helps predict potential interactions, optimize drug combinations, and avoid unexpected loss of efficacy due to lysosomal sequestration or modification.

## 4. Strategies to Exploit Lysosomal Chemistry for Cancer Therapy

Given the profound impact of lysosomes on drug disposition and cell survival, several therapeutic strategies have been developed to intentionally harness lysosomal chemistry. The approaches outlined below, some already in clinical use and others still experimental, seek to convert lysosomal barriers into opportunities for selective cancer therapy.

### 4.1. Weak-Base Trapping as a Targeting Mechanism

Although lysosomal sequestration of weak-base drugs often contributes to drug resistance, it can also be exploited for targeted delivery [38,201]. The concept involves designing lysosomotropic drugs that preferentially accumulate in cancer cells because of their higher lysosomal content or acidity [12,174]. Aggressive or drug-resistant tumor cells frequently display increased lysosomal biogenesis and acidification, particularly under metabolic stress such as hypoxia or glucose deprivation [1]. These features can enlarge the acidic compartment, allowing weakly basic cytotoxins to accumulate more extensively in tumor cells than in normal tissue.

As previously discussed in Section 3.1, chloroquine (Figure 7B) and its derivatives exemplify lysosomotropic agents. Beyond their weak-base trapping properties, these compounds have additional mechanistic relevance. By raising intralysosomal pH, chloroquine interferes with enzyme activation and autophagosome–lysosome fusion, thereby suppressing autophagy and sensitizing tumors to chemotherapy and radiotherapy [182,202]. Moreover, chloroquine-induced lysosomal alkalinization can modulate metal-dependent redox reactions and disrupt signaling pathways linked to cancer survival. Hydroxychloroquine (Figure 7C) and other analogs are currently being optimized to improve lysosomal selectivity and minimize systemic toxicity, demonstrating how pharmacological modulation of lysosomal function can complement conventional anticancer strategies.

Lysosomal targeting can also be achieved through chemical modification. Introducing weak-base functional groups, such as morpholine, directs otherwise neutral drugs to lysosomes, as morpholine preferentially accumulates in acidic environments. This approach has been used in designing fluorescent probes and could similarly guide therapeutic delivery. Such strategies may be particularly relevant for drugs acting on lysosome-associated pathways, including mammalian target of rapamycin complex 1 (mTORC1), which resides on the lysosomal membrane [205,206,207]. Targeting inhibitors to this compartment could, in principle, improve selectivity for cancer cells with hyperactive mTOR signaling.

Lysosomal pH gradients can also be utilized for tumor imaging. Certain radiotracers, including analogs of Verdazyl dyes or Amine-based probes labeled with Carbon-11 or Fluorine-18, accumulate in tumor lysosomes and generate positron emission tomography (PET) signals that correlate with lysosomal acidity or abundance [208]. By fine-tuning a compound’s pKa and lipophilicity, chemists can transform cancer cell lysosomes into drug-concentrating compartments [209]. If the accumulated agent is inherently toxic or membrane-disruptive, its preferential buildup can selectively destroy tumor cells. Achieving selectivity over normal tissues remains a challenge, but differences in perfusion, pH, and drug retention between tumor and normal tissue may provide exploitable therapeutic windows.

### 4.2. pH-Triggered Release Systems

One of the most established lysosome-exploiting strategies is pH-triggered drug release [210]. This principle underpins a range of delivery systems, including nanoparticles, liposomes, and small-molecule prodrugs. These formulations incorporate bonds or motifs that are stable at physiological pH (~7.4) but undergo cleavage or conformational change in the mildly acidic conditions (pH 5–6) of endosomes and lysosomes.

Hydrazone linkers exemplify this approach (Table 2) [186,211]. They hydrolyze rapidly under mildly acidic conditions and are widely used in polymer–drug conjugates and ADCs [212]. Although Doxorubicin HCl liposome (DOXIL) uses a different loading mechanism, the hydrazone concept remains popular for endosomal release. Acid-cleavable linkers, such as 4-(4′-acetylphenoxy)butanoate, were used in early ADCs but were replaced due to limited stability in circulation [149,213]. Modern polymeric micelles, nanogels, and liposomes frequently employ acid-sensitive linkers such as acetals, ketals, or orthoesters, which degrade selectively within the lysosomal pH range.

Calibrating the trigger pH is essential. Premature drug release can occur in the slightly acidic extracellular tumor environment (pH 6.5), whereas optimal release typically requires pH ≤ 6.0, encountered mainly in late endosomes and lysosomes. Some advanced systems employ “on-demand” activation, where acidic conditions remove protective groups to expose hydrophobic or cationic domains that then disrupt lysosomal membranes [214]. One “nanotransformer” design remains inert at pH 7 but, at pH 5, exposes a hydrophobic peptide that destabilizes lysosomal membranes and promotes cytosolic escape.

Overall, acid-sensitive linkers have become indispensable in nanomedicine, providing spatial control over drug activation [215]. Although current liposomal formulations such as DOXIL rely primarily on pH gradients for loading rather than for triggered release, future designs are expected to integrate finely tuned acid-responsive mechanisms for greater precision.

### 4.3. Redox-Responsive Drug Release and Action

Redox differences between cellular compartments can also be exploited for controlled drug release. While redox triggers are most commonly used for cytosolic targeting, where GSH concentrations are high, they can also play roles in lysosomal drug design.

Disulfide linkers are a classic example [97]. These bonds remain stable extracellularly but are cleaved by thiols such as GSH once inside cells. In some cases, disulfide-containing conjugates are first internalized into lysosomes via endocytosis and later reduced after partial lysosomal degradation or cysteine influx. Enzymes such as GILT may also participate in disulfide reduction under acidic conditions (Table 2) [94,216].

Another promising direction is ROS-responsive systems. Polymers containing peroxalate esters or arylboronic esters degrade in the presence of hydrogen peroxide, which is abundant in cancer cells and especially within lysosomes that lack catalase [217,218]. Such nanocarriers dissolve or release their payloads selectively in ROS-rich lysosomes, coupling redox activation with spatial targeting [219].

Redox-activated small molecules can also exploit lysosomal chemistry. Quinones or dihydroquinolines that undergo redox cycling may generate ROS more efficiently in the oxidative lysosomal environment. Attaching lysosome-targeting moieties to such scaffolds can focus their activity and enhance selectivity.

Although pH-triggered systems dominate current lysosome-targeted drug design, redox-responsive approaches are emerging as complementary strategies [220]. Tumor cells often display elevated ROS levels, and lysosomes exhibit distinct redox profiles compared with normal cells [221]. Combining acidic and redox-sensitive triggers within a single system offers the potential for greater specificity and control, ensuring that drug release occurs only under the unique chemical conditions of tumor lysosomes.

## 5. Enzyme-Cleavable Prodrugs and Antibody–Drug Conjugates (ADCs)

Enzyme-cleavable linkers are a cornerstone of modern targeted drug design (Table 2) [156,222]. These systems exploit lysosomal enzymes for controlled drug activation, transforming what was once a degradative process into a therapeutic advantage.

### 5.1. Antibody–Drug Conjugates

Nearly all ADCs depend on endosomal or lysosomal processing to release their cytotoxic payloads [223]. The most established example involves cathepsin-B-cleavable dipeptide linkers such as Val-Cit and Val-Ala, which have been incorporated into several FDA-approved ADCs [147,224]. Other enzyme-responsive systems have also been explored. For instance, legumain-cleavable linkers were developed for experimental seco-CBI conjugates, a class of DNA-alkylating ADC payloads. Legumain, an asparaginyl endopeptidase enriched in tumor lysosomes, recognizes a specific peptide motif in the linker, ensuring that drug release occurs selectively in cells with high legumain activity.

Another design utilizes β-glucuronide linkers, as seen in glucuronide derivatives of Doxorubicin [225]. In this case, β-glucuronidase, released from the lysosomes of dying tumor cells or from tumor-associated stroma, cleaves the linker to liberate the active drug in the tumor microenvironment. Although the cleavage may not always occur within intact lysosomes, the underlying principle of enzyme-triggered activation remains consistent across these systems.

### 5.2. Peptide- and Polymer–Drug Conjugates: Enzyme-Cleavable Systems and Design Considerations

Smaller analogs of antibody–drug conjugates (ADCs), known as peptide–drug conjugates (PDCs), employ tumor-homing peptides instead of antibodies to achieve targeted delivery [226]. Following receptor-mediated endocytosis, lysosomal proteases cleave the linker to release the active payload. For example, the cytolytic peptide melittin has been linked to a matrix metalloproteinase (MMP)-sensitive sequence to enable activation specifically in MMP-rich tumor environments [227]. Although MMPs often act extracellularly, similar strategies can be applied to lysosomal proteases such as cathepsin D, which functions optimally in the acidic lysosomal lumen and can trigger intracellular activation.

Synthetic polymer–drug conjugates also employ enzyme-responsive linkers for selective release within lysosomes [228]. Notably, N-(2-hydroxypropyl)-methacrylamide (HPMA) copolymers use Gly–Phe–Leu–Gly spacers that are cleaved by cathepsin B [229]. These systems mirror ADC linker design principles but replace the antibody with a synthetic polymer scaffold, providing greater versatility in molecular weight, composition, and pharmacokinetic control.

Enzyme-cleavable conjugates rely critically on the activity of specific lysosomal enzymes, which can vary among tumor types and microenvironments due to differences in pH, oxygen tension, and metabolic stress [156]. While cathepsins are generally active under acidic conditions, tumor hypoxia or lysosomal rupture may affect their localization or catalytic efficiency. Off-target activation in normal tissues remains a potential limitation; however, ADCs and PDCs maintain selectivity primarily through receptor- or antigen-mediated endocytosis, minimizing systemic exposure. Collectively, these enzyme-cleavable systems exemplify how lysosomal protease activity can be harnessed for precise intracellular drug activation.

## 6. Emerging Concepts in Lysosomal Modulation and Targeted Chimeras

An exciting frontier in drug design involves redirecting or exploiting lysosomal machinery for therapeutic benefit. Among these strategies, lysosome-targeting chimeras (LYTACs) have emerged as bifunctional molecules capable of recruiting extracellular or membrane proteins to lysosomes for degradation through receptors such as the asialoglycoprotein receptor [230,231,232,233]. Conceptually, LYTACs are the lysosomal counterparts of proteolysis-targeting chimeras (PROTACs) that drive proteasomal degradation [234]. Although still early in development, LYTACs represent a powerful way to harness the lysosomal degradation pathway for eliminating oncogenic or pathogenic membrane proteins.

Harnessing lysosomal enzymes for controlled drug release is now validated by several approved therapeutics and many experimental candidates. This approach leverages the abundance and catalytic efficiency of lysosomal hydrolases, which remain sequestered from the extracellular space, thereby providing intrinsic selectivity. Future work will likely refine enzyme-cleavable linkers to improve tissue specificity, activation kinetics, and stability, further consolidating lysosomal enzymology as a foundation for precision drug delivery.

### 6.1. Overcoming Lysosomal Drug Sequestration and Combination Strategies

An emerging theme in cancer pharmacology is to exploit the very mechanisms that confer drug resistance. Lysosomal sequestration, once viewed purely as a liability, is now being reimagined as a therapeutic opportunity. Multiple strategies are being developed to counter or repurpose this process, including combination therapies that release trapped agents and Pgp-targeted approaches that redirect drug transport to sensitize cancer cells.

#### 6.1.1. Combination Therapy to Release Trapped Drugs

A powerful example of this approach is the combination of classical chemotherapeutics, such as Doxorubicin, with lysosome-disrupting agents like thiosemicarbazones, Dp44mT and DpC [9,178]. The concept is to allow resistant tumor cells to sequester the chemotherapeutic drug in lysosomes via Pgp activity and pH trapping and then introduce a second agent that causes LMP [9].

Seebacher et al. demonstrated that Pgp-expressing tumor cells, normally unresponsive to Doxorubicin, were efficiently killed when treated with Dp44mT or DpC (Figure 9A,B) [9,178]. These thiosemicarbazones localize to lysosomes and generate ROS, triggering LMP and releasing sequestered Doxorubicin into the cytosol (Figure 10). The resulting synergy was strongly Pgp dependent: inhibition or knockdown of Pgp abolished the effect. In essence, the efflux pump was repurposed to deliver drugs into lysosomes rather than expel them from the cell [9].

Because both Dp44mT and DpC (Figure 9A,B) are Pgp substrates, they preferentially accumulate in Pgp-high cells and their lysosomes [9]. This creates a form of self-selectivity: resistant cells that overexpress Pgp accumulate both drugs and the lysosomal disruptor, whereas Pgp-low cells, already sensitive to Doxorubicin, experience minimal exposure to DpC. The result is selective destruction of resistant cells. Clinically, both DpC and COTI-2 have entered trials as single agents (Table 3), with preclinical studies indicating that DpC in particular shows strong synergy with Doxorubicin and related chemotherapeutics [235]. This combination represents a compelling proof-of-principle for “smart combination therapy,” where a resistance mechanism is deliberately exploited to restore chemosensitivity.

#### 6.1.2. Acridine–Thiosemicarbazone Hybrids: Targeting Lysosomes by Design

Building upon this concept, N-acridine thiosemicarbazones (NATs; Figure 11A,B) were developed to integrate a Pgp-avid acridine moiety with a metal-binding thiosemicarbazone warhead [8]. Acridine, a well-known DNA intercalator and lysosomotropic fluorophore, confers both fluorescence and lysosomal targeting [8]. In Pgp-expressing cells, NATs are actively transported into lysosomes, where they undergo redox cycling with Fe(III) or Cu(II), generating ROS that disrupt the lysosomal membrane (Figure 11A,B) [8]. In contrast, in Pgp-negative cells, partial nuclear localization occurs due to acridine’s affinity for DNA. This dual localization pattern correlates with dual mechanisms of cytotoxicity: lysosomal rupture in resistant cells and DNA intercalation in sensitive cells.

Fluorescence microscopy confirmed that NATs co-localize with lysosomal markers in resistant cells, while Pgp inhibition abolished both lysosomal accumulation and cytotoxicity. The result is an elegant inversion of drug resistance: NATs use Pgp to reach their site of action, effectively “riding the pump” to the lysosome to induce cell death (Figure 11A,B). Their intrinsic fluorescence also allows real-time imaging, making them promising theranostic agents that couple therapeutic and diagnostic functions [8].

#### 6.1.3. Lysosome-Targeted Photodynamic and Photothermal Therapy

Another method to exploit lysosomal sequestration involves photosensitizers or plasmonic nanoparticles that accumulate in lysosomes and can be externally activated [236]. Upon irradiation with light or exposure to a magnetic field, these agents produce localized ROS or heat, leading to rapid LMP and cell death.

For example, morpholine-bearing chlorine derivatives accumulate in lysosomes and, when illuminated, generate singlet oxygen that causes lysosomal rupture and necrotic cell death [237]. Similarly, gold nanoparticles localized to lysosomes can be activated by laser irradiation to induce photothermal ablation, selectively damaging tumor cells while sparing normal tissue. Such externally triggered systems can bypass multidrug resistance entirely, since they depend on physical activation rather than continuous drug accumulation.

#### 6.1.4. Targeting Lysosomal Membrane Proteins

Beyond Pgp, other lysosomal membrane proteins offer potential targeting opportunities. Lysosomal-associated membrane protein 1 (LAMP-1) and Lysosomal-associated membrane protein 2 (LAMP-2), for instance, are highly glycosylated proteins that are often overexpressed on the surface of invasive tumor cells [159]. ADCs directed against these proteins could, in theory, deliver cytotoxic payloads directly to lysosomes after internalization [97]. Another emerging target is proline–glutamine loop containing 2 (PQLC2), a lysosomal lysine/arginine transporter that is upregulated in certain cancers [238]. These membrane proteins may serve as selective entry points for lysosome-targeted therapeutics in future designs.

#### 6.1.5. Sensitizing Cells to Ferroptosis via Lysosomal Iron

CD44-high, mesenchymal-like tumor cells harbor large lysosomal iron stores that can be therapeutically exploited [239]. Strategies include delivering iron-binding compounds that liberate reactive iron within lysosomes or employing iron-oxide nanoparticles that dissolve under acidic conditions to fuel localized ROS generation. These interventions shift the lysosomal milieu toward a pro-ferroptotic state, amplifying oxidative damage and triggering selective death of iron-rich cancer cells.

Collectively, these strategies represent a paradigm shift in cancer drug design: lysosomal sequestration is no longer viewed merely as a passive sink that diminishes efficacy but as an active, targetable process. By leveraging lysosomal accumulation, redox chemistry, and membrane dynamics, researchers are transforming a classical resistance mechanism into a therapeutic frontier.

### 6.2. Metal-Based Drugs and Lysosomal Redox Mechanisms

Metal-based therapeutics offer rich opportunities for exploiting the redox-active environment of lysosomes [240]. Beyond classical metal-chelating drugs such as thiosemicarbazones, a diverse range of metal complexes can either utilize or perturb lysosomal chemistry to trigger selective cytotoxicity.

#### 6.2.1. Copper Complexes

Several copper complexes of small ligands exhibit potent cytotoxicity and preferential lysosomal accumulation [24,27]. For instance, Triapine, an iron chelator evaluated in clinical trials, can also bind copper, and its Cu complex has been observed to localize within lysosomes, contributing to oxidative damage [241]. The Dp44mT (Figure 9A) family and its analogs form Cu(II) complexes that redox cycle within lysosomes, generating ROS and inducing LMP [24,27].

Recent insights into cuproptosis, a copper-dependent form of cell death involving mitochondrial ferredoxin (FDX1) and lipoylated enzymes, raise questions about the potential role of lysosomes as upstream contributors [242]. Lysosomes act as key copper storage sites, and compounds that trigger copper release from these compartments may amplify intracellular toxicity through oxidative and proteotoxic stress [243].

#### 6.2.2. Gold Compounds

Gold-based drugs primarily target thiol-containing enzymes such as thioredoxin reductase in the cytosol and mitochondria, yet some gold complexes are specifically designed to act in lysosomes [244,245]. Gold(III) complexes are relatively stable under acidic, oxidizing conditions, making lysosomes a favorable environment for their persistence and activation.

For example, gold(III)–porphyrin complexes remain intact in neutral conditions but can be reduced to gold(I) in lysosomes by local reductants such as glutathione or cysteine [246]. The resulting gold(I) species irreversibly bind to cysteine proteases and other thiol-containing proteins, leading to enzyme inactivation and ROS generation. Cationic gold(I)–phosphine or N-heterocyclic carbene complexes also tend to accumulate in lysosomes, where their heavy metal toxicity causes lysosomal swelling and rupture. These mechanisms highlight how tuning redox potential and charge distribution allows gold complexes to harness the unique chemistry of the lysosomal lumen.

#### 6.2.3. Ferroptosis Inhibitors and Radical-Trapping Antioxidants

In contrast to pro-oxidant metal pathways, ferroptosis inhibitors can suppress lysosomal and lipid ROS production and protect cells from oxidative damage [82]. Compounds such as liproxstatin-1 (Figure 12), which function as potent lipid radical-trapping antioxidants (RTAs), intercept lipid peroxyl radicals and halt the propagation of lipid peroxidation [247,248]. Rather than chelating iron, liproxstatin-1 stabilizes vulnerable membranes and prevents iron-driven oxidative injury.

This functional duality offers design flexibility: ferroptosis inhibitors can shield normal tissues during oxidative therapies, or conversely, be withheld in cancer contexts to amplify iron-dependent cytotoxicity. Strategic combinations of ROS-inducing agents and ferroptosis modulators may therefore fine-tune the balance between protection and destruction across distinct cell populations.

#### 6.2.4. Photodynamic Metal Complexes

Some ruthenium, zinc, or phthalocyanine-based complexes act as lysosome-localized photosensitizers [249]. Upon light activation, they generate singlet oxygen within lysosomes, causing localized oxidative bursts and LMP. This approach combines metal redox activity with spatially controlled photodynamic or photothermal therapy, providing high precision in tumor cell destruction.

## 7. Future Perspectives

Viewing the lysosome as a chemical reactor opens new possibilities for cancer therapy. Future research will focus on improving tumor selectivity, overcoming drug resistance, and translating lysosomal targeting into safe and effective treatments.

Selective targeting will depend on identifying features that distinguish cancer lysosomes from those in normal cells, such as altered acidity, membrane proteins, or metal content. Designing ligands, peptides, or nanocarriers that recognize these traits could achieve precise drug delivery to tumor lysosomes.

Because lysosomal sequestration contributes to resistance, combination strategies that adjust lysosomal pH or disrupt its membrane can restore sensitivity to therapy. Agents such as DpC and Doxorubicin already illustrate this potential. Similarly, activating lysosome-dependent death pathways such as ferroptosis or controlled membrane rupture may provide alternatives for apoptosis-resistant tumors. Advances in imaging and molecular profiling will allow mapping of lysosomal activity across cancers, supporting more personalized drug design. Smart carriers that respond to pH, redox state, or enzyme activity will enable precise intracellular release. Biomarkers of lysosomal function, including plasma cathepsins or imaging tracers, can help monitor safety.

Progress in this field will rely on collaboration between chemists, biologists, pharmacologists, and clinicians. Together, these efforts will transform the lysosome from a degradative organelle into a controllable site for selective drug activation and mechanism-based cancer therapy.

## 8. Conclusions

The lysosome has emerged from its traditional view as a degradative compartment to become a central player in determining drug fate and therapeutic response [157]. Recognizing it as a chemical reactor reframes how we design and deploy anticancer agents. Over the past decade, research has moved beyond the autophagy paradigm to reveal lysosomes as chemical battlegrounds that dictate whether drugs succeed or fail. Future therapies will increasingly incorporate this understanding, developing ways to subvert or weaponize the lysosomal environment for selective tumor destruction. The challenge ahead lies in safely harnessing this potent organelle, but the potential rewards, more effective treatments for resistant cancers, make this one of the most promising frontiers in modern drug discovery.

## Figures and Tables

**Figure 1 ijms-26-11581-f001:**
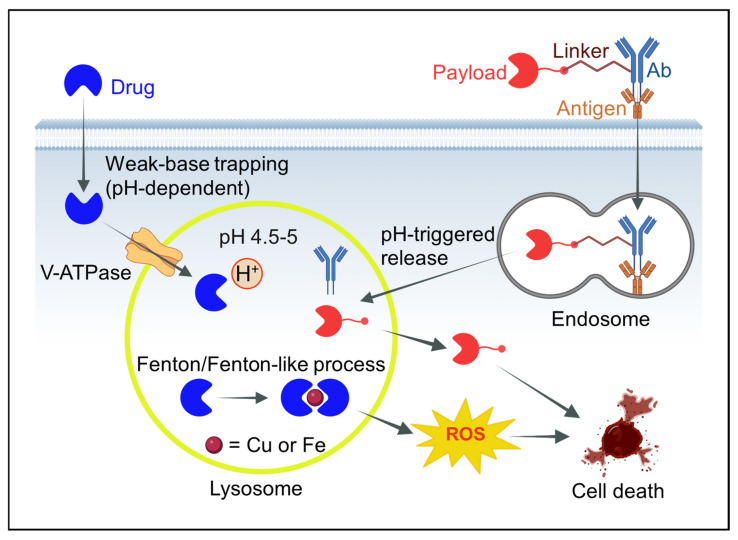
Integrated model of lysosomal chemistry governing anticancer drug fate. Weak-base drugs enter cells and accumulate in lysosomes through pH-dependent trapping driven by the V-ATPase proton pump. The acidic lumen (pH 4.5–5) promotes protonation of weak bases and provides favorable conditions for Fenton and Fenton-like reactions involving labile Fe or Cu, which generate reactive oxygen species (ROS). Antibody–drug conjugates (Ab–linker–payload) are internalized via endocytosis and encounter progressive acidification along the endosomal–lysosomal pathway, facilitating cleavage of acid-labile linkers and controlled release of the cytotoxic payload. Together, pH, metal-driven redox chemistry, and pH- or enzyme-triggered bond cleavage converge to regulate drug sequestration, activation, efficacy, and lysosome-dependent cell death. Created in BioRender. Dharmasivam, M. (2025) https://app.biorender.com/68e337b54289dc9ed50e4581 (accessed on 12 October 2025).

**Figure 2 ijms-26-11581-f002:**
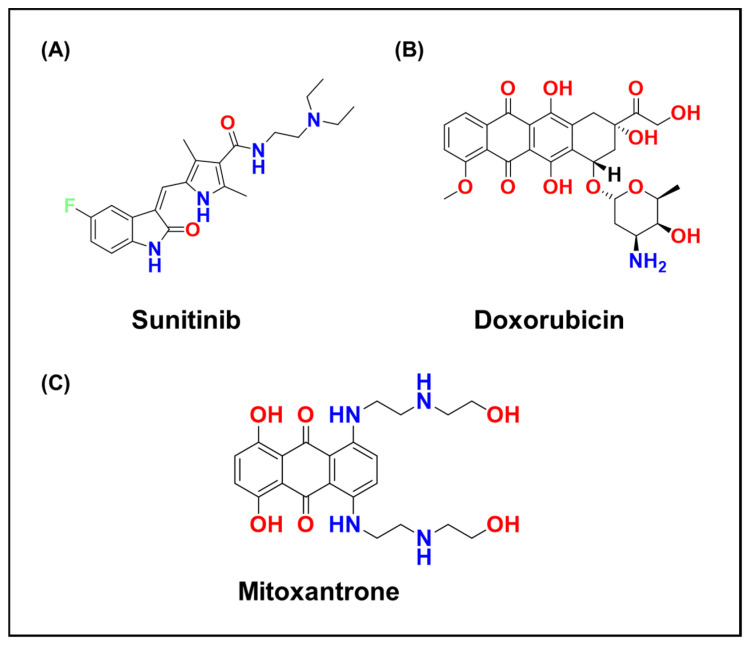
Chemical structures of (**A**) Sunitinib, (**B**) Doxorubicin, and (**C**) Mitoxantrone.

**Figure 3 ijms-26-11581-f003:**
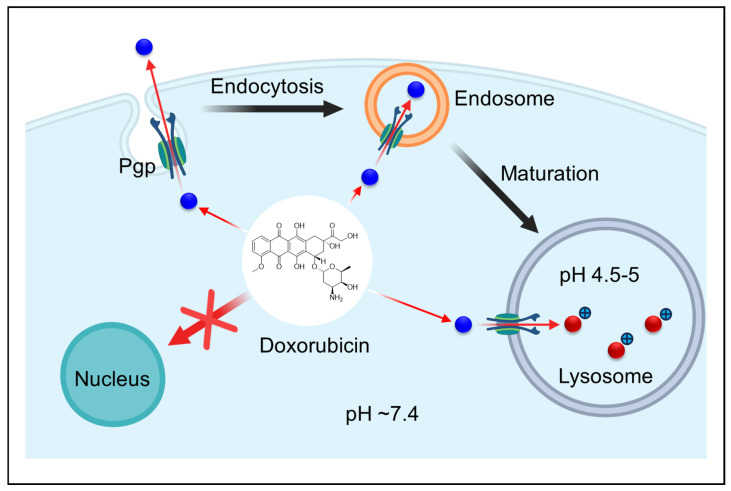
P-glycoprotein-mediated lysosomal sequestration of doxorubicin. Doxorubicin, a cationic anthracycline, is effluxed by Pgp transporters from the cytosol and redirected into the endosomal–lysosomal pathway through endocytosis. As endosomes mature into lysosomes (pH 4.5–5), the acidic environment promotes protonation and trapping of Doxorubicin, preventing its accumulation in the nucleus where it normally intercalates with DNA. This lysosomal sequestration contributes to multidrug resistance by reducing the cytotoxic availability of doxorubicin at its nuclear target. Created in BioRender. Dharmasivam, M. (2025) https://app.biorender.com/68e4aa8d75752856e7baefbe (accessed on 10 October 2025).

**Figure 4 ijms-26-11581-f004:**
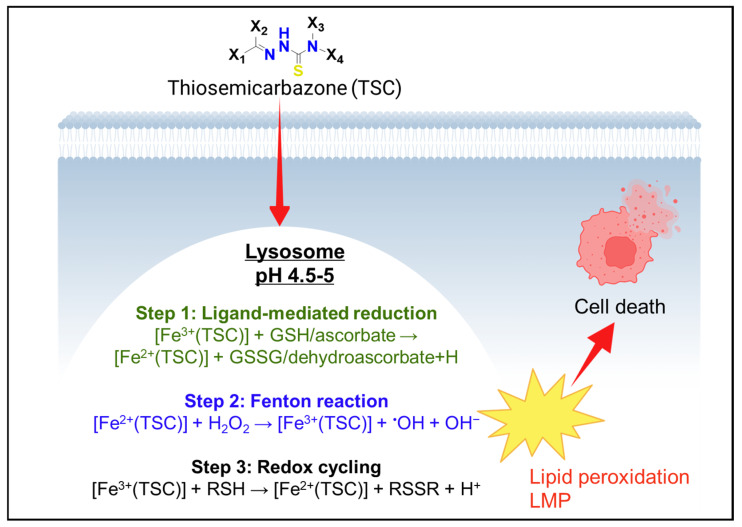
Redox cycling of thiosemicarbazones (TSCs) within the lysosome. TSCs accumulate in lysosomes (pH 4.5–5), where [Fe^3+^(TSC)] complexes are reduced by cellular reductants such as GSH or ascorbate (**Step 1**). The resulting [Fe^2+^(TSC)] species catalyze Fenton reactions with H_2_O_2_ to generate hydroxyl radicals (**Step 2**), which promote lipid peroxidation and LMP. Continuous Fe^2+^/Fe^3+^ interconversion (**Step 3**) sustains ROS production, culminating in cell death. Created in BioRender. Dharmasivam, M. (2025) https://app.biorender.com/68e9c3d7d0a4e2349c63e61a (accessed on 11 October 2025).

**Figure 5 ijms-26-11581-f005:**
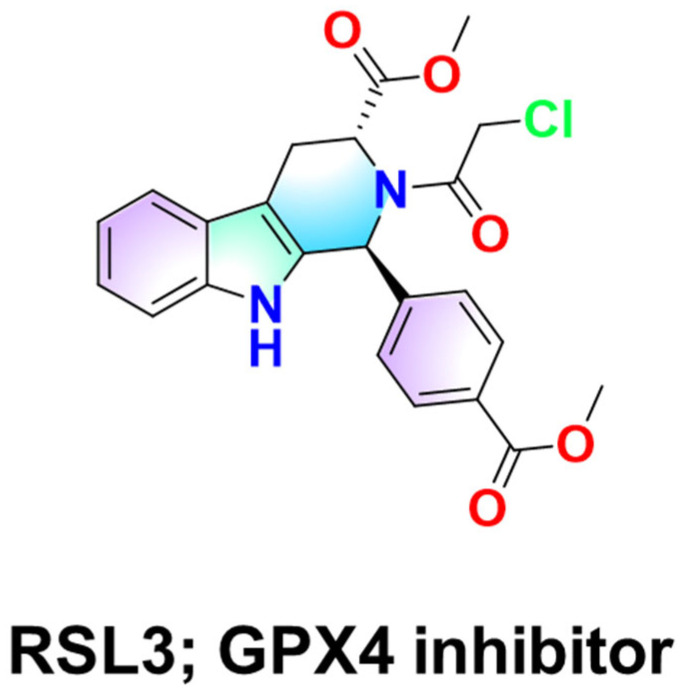
Chemical structure of RSL3, a covalent inhibitor of glutathione peroxidase 4 (GPX4). RSL3 inhibits GPX4 by covalently binding to the enzyme’s selenocysteine residue, leading to depletion of lipid peroxide detoxification capacity. This inhibition triggers ferroptosis through the accumulation of lipid hydroperoxides and oxidative damage to cellular membranes.

**Figure 6 ijms-26-11581-f006:**
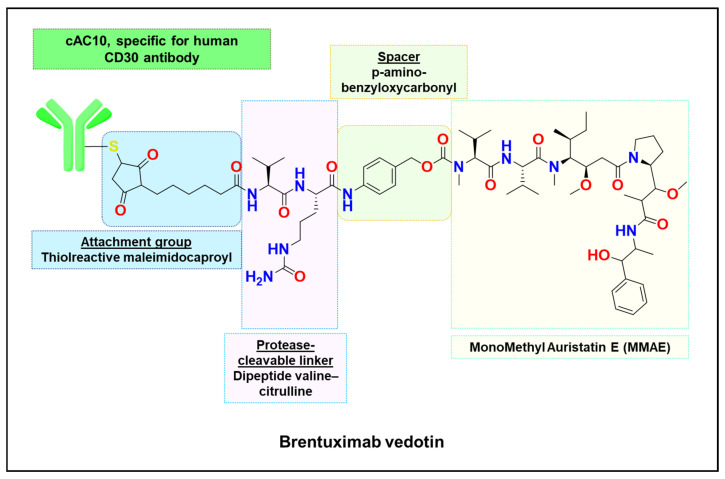
Structural composition of brentuximab vedotin, an antibody–drug conjugate (ADC) targeting CD30-positive cancers. Brentuximab vedotin consists of the cAC10 monoclonal antibody specific for the human CD30 antigen, covalently linked to the cytotoxic payload monomethyl auristatin E (MMAE) via a thiol-reactive maleimidocaproyl attachment group, a protease-cleavable valine–citrulline dipeptide linker, and a *p*-aminobenzylcarbamate (PABC) spacer. Upon internalization into CD30-expressing tumor cells, lysosomal proteases cleave the linker to release MMAE, which disrupts microtubule assembly and induces apoptotic cell death.

**Figure 7 ijms-26-11581-f007:**
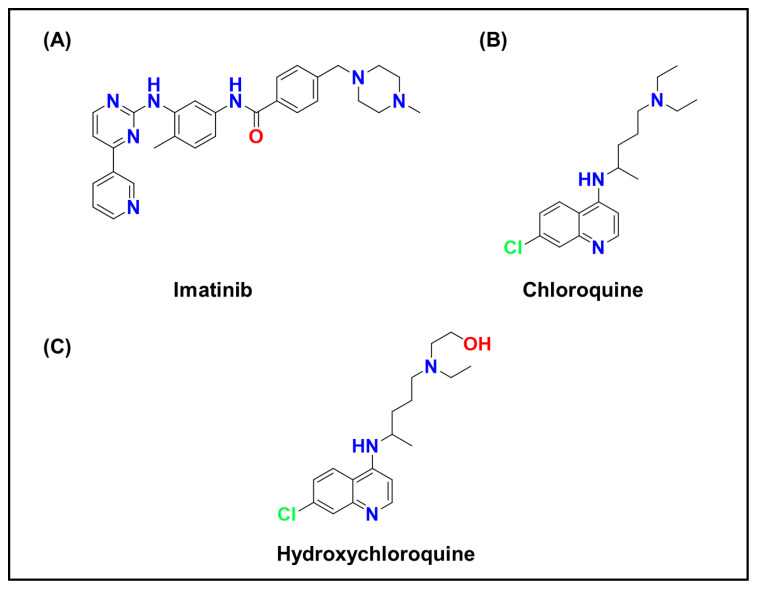
Chemical structures of (**A**) Imatinib, (**B**) Chloroquine, and (**C**) Hydroxychloroquine. Imatinib is a selective tyrosine kinase inhibitor that targets BCR-ABL, c-KIT, and PDGFR kinases, widely used in the treatment of chronic myeloid leukemia and gastrointestinal stromal tumors. Chloroquine and its hydroxylated analog hydroxychloroquine are weakly basic lysosomotropic agents that accumulate in acidic organelles, elevate lysosomal pH, and inhibit autophagic flux. These compounds are commonly employed as pharmacological tools to investigate lysosomal function, autophagy, and intracellular redox homeostasis.

**Figure 8 ijms-26-11581-f008:**
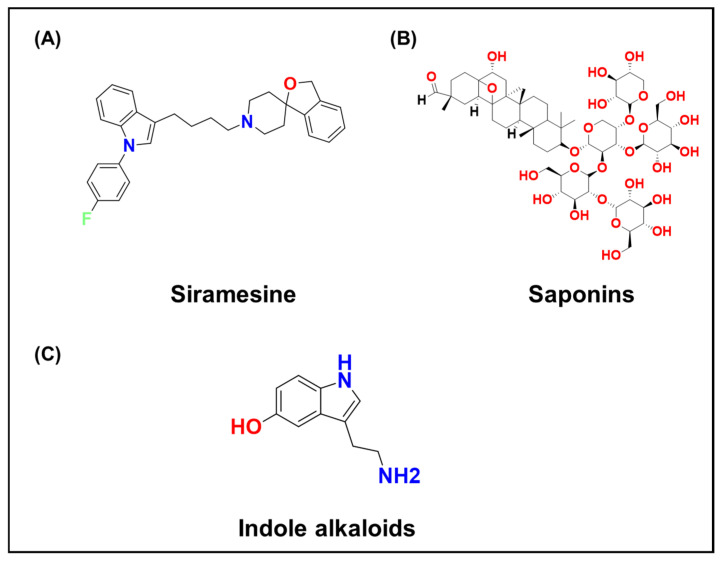
Representative lysosomotropic and membrane-active agents. (**A**) Siramesine, a sigma-2 receptor agonist, induces lysosomal membrane permeabilization and cell death through disruption of lysosomal integrity. (**B**) Saponins, a class of amphiphilic glycosides composed of a hydrophobic triterpenoid or steroidal backbone linked to multiple sugar moieties, interact with membrane sterols to enhance permeability and facilitate lysosomal destabilization. (**C**) Indole alkaloids, exemplified by tryptamine derivatives, contain a characteristic indole ring that mediates redox activity and lysosomal accumulation, contributing to their biological and pharmacological effects.

**Figure 9 ijms-26-11581-f009:**
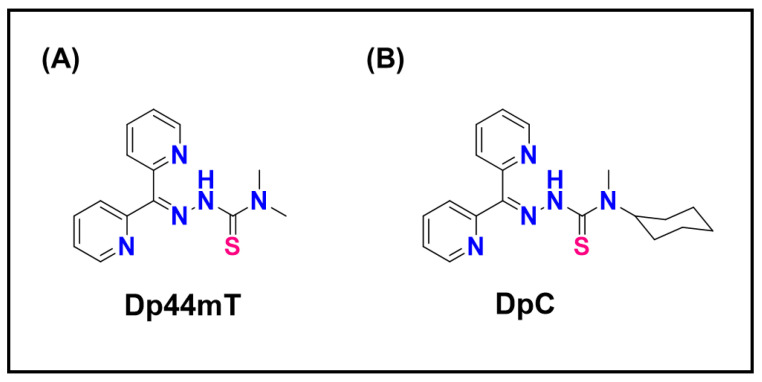
Chemical structures of lysosomotropic thiosemicarbazone analogs (**A**) Dp44mT and (**B**) DpC both localize to lysosomes, where they undergo redox cycling with iron to ROS, promoting lysosomal membrane permeabilization and cell death. The structural modification in DpC, featuring a bulky cyclohexyl substituent, enhances its metabolic stability and safety profile while retaining potent lysosomal targeting and anticancer activity.

**Figure 10 ijms-26-11581-f010:**
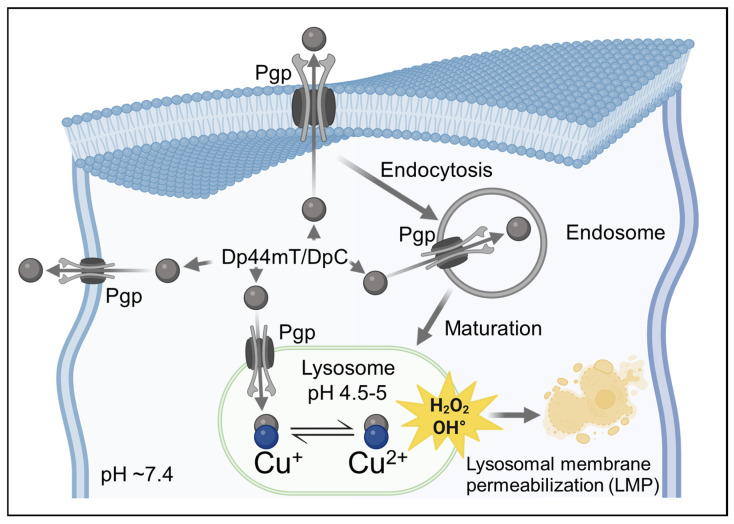
Pgp-mediated lysosomal transport and redox cycling of thiosemicarbazones. The lysosomotropic thiosemicarbazones Dp44mT and DpC are transported by Pgp from the cytosol into the endosomal–lysosomal pathway through endocytosis. As endosomes mature into lysosomes (pH 4.5–5), these complexes accumulate and undergo redox cycling between Cu^2+^ and Cu^+^, generating ROS such as hydroxyl radicals (•OH) and H_2_O_2_. The accumulation of ROS triggers LMP, leading to oxidative damage and ultimately cell death. Created in BioRender. Dharmasivam, M. (2025) https://app.biorender.com/68e915e070d05657446c429c (accessed on 11 October 2025).

**Figure 11 ijms-26-11581-f011:**
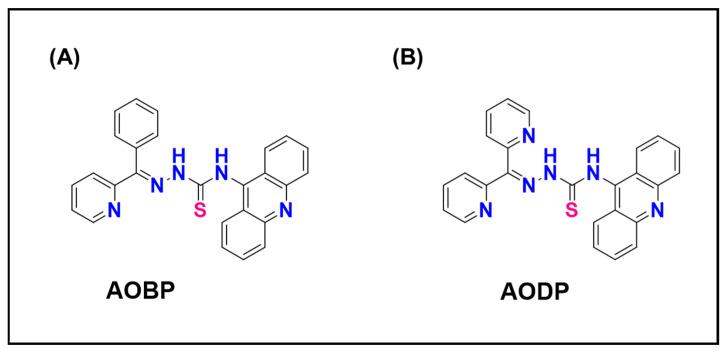
Chemical structures of acridine-containing thiosemicarbazone analogs (**A**) AOBP and (**B**) AODP incorporate an extended π-conjugated acridine system (highlighted) that enhances lysosomal localization, fluorescence tracking, and DNA intercalation potential. These structural modifications increase lipophilicity and cellular uptake while maintaining strong metal-binding and redox-active thiosemicarbazone cores.

**Figure 12 ijms-26-11581-f012:**
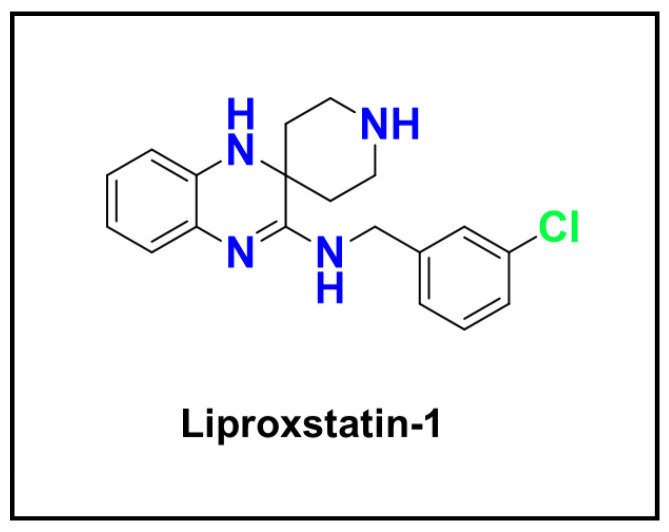
Chemical structure of liproxstatin-1, a potent ferroptosis inhibitor. Liproxstatin-1 is a small-molecule antioxidant that prevents ferroptotic cell death by inhibiting lipid peroxidation. It functions as a radical-trapping agent and preserves membrane integrity through the suppression of reactive oxygen species (ROS)-driven lipid oxidation, thereby maintaining redox homeostasis under oxidative stress conditions.

**Table 1 ijms-26-11581-t001:** Key Chemical Features of the Lysosome.

Parameter	Chemical Basis	Biological/Pharmacological Implication
pH (~4.5–5.0)	Maintained by V-ATPase proton pump [62].	Promotes weak-base drug trapping and hydrolysis of acid-labile bonds [63].
Redox Potential	Oxidizing environment with Fe^2+^/Cu^+^ cycling and low GSH	Enables Fenton chemistry; supports ROS-driven damage and ferroptosis [64].
Thiol Reductase (GILT)	Cys–His–Asp catalytic triad active at pH ~5	Catalyzes disulfide reduction; crucial for MHC class II antigen processing [65].
Hydrolases	>60 enzymes (cathepsins, lipases, phosphatases, etc.)	Catalyze degradation or prodrug activation (e.g., ADCs) [66].
Metal Ions (Fe, Cu, Zn)	Accumulate via autophagy and protein turnover	Drive redox reactions, ferroptosis, and chemodynamic therapy [67].
Reactive Oxygen Species	H_2_O_2_ and •OH generated via Fenton/Haber–Weiss	Cause lipid peroxidation and lysosomal membrane permeabilization (LMP) [68].

**Table 2 ijms-26-11581-t002:** Drug Design and Therapeutic Strategies Exploiting Lysosomal Chemistry. This table summarizes representative approaches that utilize lysosomal properties for therapeutic advantage. The term Therapeutic + Diagnostic (Theranostic) Approach refers to dual-function agents that combine therapeutic and diagnostic capabilities within a single molecular or nanostructured system, enabling simultaneous drug delivery, imaging, and treatment monitoring.

Strategy	Mechanism	Representative Example	Outcome
pH-Triggered Release	Acid-labile linkers (hydrazone, acetal) cleave at pH ≤ 6	Hydrazone-linked doxorubicin nanocarriers	Controlled drug release in lysosomes
Redox-Responsive Systems	Disulfide or ROS-sensitive bonds activated by GILT or H_2_O_2_	Disulfide-linked prodrugs, peroxalate esters	Selective activation in oxidative lysosomes
Enzyme-Cleavable Linkers	Cathepsin or β-glucuronidase-mediated cleavage	ADCs with Val–Cit linker	Site-specific payload release
Weak-Base Trapping	Exploiting tumor lysosomal acidity and volume	Chloroquine, acridine hybrids	Preferential tumor accumulation
LMP-Inducing Combinations	Trigger lysosomal rupture to release trapped drugs	DpC + Doxorubicin	Overcomes Pgp-mediated resistance
Metal-Based Lysosomal Drugs	Redox-active metal complexes generate ROS	Cu(II)– or Au(III)–thiosemicarbazones	Induce lysosomal oxidative damage
Therapeutic + Diagnostic (Theranostic) Approach	Fluorescent or radiolabeled lysosomotropic drugs	Acridine–TSC hybrids, LysoRhoNox-M	Real-time tracking and dual therapy

**Table 3 ijms-26-11581-t003:** Representative ongoing or recent clinical trials of lysosome-targeting or lysosomotropic anticancer agents.

Compound	Cancer Type/Population	Intervention Description	Trial Phase	Status	Clinical Identifier/ Reference
Chloroquine (CQ)	Glioblastoma, pancreatic, and breast cancers	CQ as autophagy/lysosome inhibitor in combination with chemotherapy or radiotherapy	Phase II–III	Active/ completed	NCT02378532, NCT01446016
Hydroxychloroquine (HCQ)	Pancreatic and lung cancers	HCQ used to inhibit autophagy and enhance chemotherapy response	Phase I–II	Active/ completed	NCT01506973, NCT01273805
Triapine	Cervical, ovarian, and hematologic malignancies	Ribonucleotide-reductase inhibitor with redox-active metal coordination and lysosomal accumulation	Phase II	Active/ recruiting	NCT02466971
DpC	Advanced and drug-resistant solid tumors	Lysosomotropic metal-binding thiosemicarbazone inducing ROS via redox cycling	Phase I (NCT02688101)	Completed	NCT02688101
COTI-2	Head and neck, gynecologic, and brain tumors	Thiosemicarbazone analog targeting mutant p53 and lysosomal pathways	Phase I	Active	NCT02433626

## Data Availability

No new data were created or analyzed in this study. Data sharing is not applicable to this article.

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
