# Peer review of "Lysosome as a Chemical Reactor"

_ijms, 2025, doi:10.3390/ijms262311581_

Round 1

Reviewer 1 Report

Comments and Suggestions for Authors

The manuscript covers a very interesting topic, i.e. the multifaceted role of lysosomes, with an original and innovative perspective, i.e. their role in drug behaviour from different point of views. It is generally well organized, but some parts would benefit from a better organization

Major points

In the Abstract, the sentence “It has emerged as a chemically dynamic organelle that functions as a microenvironmental reactor shaping drug behaviour in cancer cells.” is out of context, as many interesting features of lysosomes have emerged beyond degradation (metabolism, signalling), as correctly listed in the Introduction. The sentence should be re-formulated

Line 78 The sentence “…key chemical parameters of the lysosomal lumen, including acidity, redox balance, thiol pools, enzymatic content, and metal ions…” appears just a list and are not further critically discussed. Could authors discuss better these factors, whether they rely on each other, or whether there a hierarchy among them?

Line 145-149 Please add further details, i.e. one-two brief sentences reporting a few examples of drugs activated and/or inactivated by acidic pH

Line 182 Could authors provide a reference for the sentence “These ROS oxidize lipids, proteins, and nucleic acids in proximity to the lysosomal membrane.”, or it is just a speculation?

Line 185 The sentence “Recent studies have shown that the ferroptosis inducer RAS-selective lethal 3 (RSL3; Figure 5) promotes lipid oxidation specifically within lysosomes, whereas lipophilic antioxidants that localize to lysosomes can suppress ferroptosis by inactivating intralysosomal iron [82, 84].” is unclear and should be re-written

Line 321 Authors wrote that “These iron-rich lysosomes can act as potential “time bombs” for Fenton chemistry [120]”. However, it is unclear how this phenomenon ca help cancer cell survival and spreading. Coud authors clarify it?

In section 3, the subsection 3.4 appears the logical consequence of subsection 3.2. Subsection 3.3. should be moved at the end of section 3.

In section 5.1, the example of chloroquine is redundant. The behaviour of chloroquine as a weak base was introduced in the previous sessions, with examples of its possible use in combination with other drugs. The session should remind to previous sections and illustrate additional original content on chloroquine, if any.

In Table 2, what do authors mean with the definition “Theranostics” as Strategy? It is unclear

Sections 6.2 to 6.4: The organization in such small subsections can be avoided, as it increase the overall length without improving the clarity

The organization of section 7 is awkward: the title is “Emerging Concepts: Lysosome-Targeting Chimeras”, but the subsection 7.1 title is “Overcoming Lysosomal Drug Sequestration: P-gp Repurposing and Lysosome-Targeted 728 Combinations”.

Section 7.1.5 begins with “Cancer cells with elevated lysosomal iron are particularly susceptible to ferroptosis, a form of cell death driven by iron-dependent lipid peroxidation. This is redundant, as it has been already explained in the text

The section “Future perspectives” is too long and fragmented: it should more concise and focused

Author Response

14th November 2025

Referee 1: Recommendation: Major revisions and stated: “The manuscript covers a very interesting topic, i.e. the multifaceted role of lysosomes, with an original and innovative perspective, i.e. their role in drug behaviour from different point of views. It is generally well organized, but some parts would benefit from a better organization.”

We thank the reviewers for their very constructive and insightful feedback. We appreciate the opportunity to revise our manuscript and believe that the revisions, along with the addition of the new section, and table have substantially enhanced its clarity and impact.

We are also grateful to you and the reviewers for your thoughtful evaluation. We are confident that the revised version satisfactorily addresses all concerns raised and will be of significant interest to the readers of International Journal of Molecular Sciences.

Kind regards,

Dr. Mahendiran Dharmasivam, B.Sc., M.Sc., B.Ed., Ph.D., MRACI

National Breast Cancer Foundation (NBCF) Fellow

Institute for Biomedicine and Glycomics, Griffith University Gold Coast

Southport QLD 4215, Australia

Phone: +61-7555-27502; Email: m.dharmasivam@griffith.edu.au

Referee 1: Recommendation: Major Revision

 Q1. In the Abstract, the sentence “It has emerged as a chemically dynamic organelle that functions as a microenvironmental reactor shaping drug behaviour in cancer cells.” is out of context, as many interesting features of lysosomes have emerged beyond degradation (metabolism, signalling), as correctly listed in the Introduction. The sentence should be re-formulated.

Revision: Thank you, and we agree. The original phrasing could imply exclusivity and underplay the broader lysosomal functions (degradation, metabolism, and signalling) that are already discussed in the Introduction. We have revised the sentence to retain the concept of the lysosome as a “chemical microreactor” while explicitly situating it alongside these established roles.

Revised abstract sentence (page 1):

The lysosome is not only degradative; its acidic, redox-active lumen also serves as a chemical ‘microreactor’ that can modulate anticancer drug disposition and activation.

Q2. Line 78 The sentence “…key chemical parameters of the lysosomal lumen, including acidity, redox balance, thiol pools, enzymatic content, and metal ions…” appears just a list and are not further critically discussed. Could authors discuss better these factors, whether they rely on each other, or whether there a hierarchy among them?

Revision: Thank you for this valuable suggestion. We agree that the listed parameters required deeper analysis rather than a simple enumeration. Accordingly, we have revised the relevant section to provide a more integrated and critical discussion of how these factors interact and influence one another within the lysosomal microenvironment.

Specifically, we now highlight that acidity acts as the primary driver establishing the lysosomal chemical hierarchy, governing enzyme activity, metal ion solubility, and redox equilibria. The redox balance, in turn, modulates thiol pools and reactive oxygen species chemistry, which together determine the oxidative or reductive reactivity of the lumen. These parameters collectively influence enzymatic function and metal-catalysed transformations, meaning that perturbations in one (for example, pH elevation or GSH depletion) can propagate across others to reshape lysosomal chemistry and drug fate.

The following paragraphs have been added to the revised manuscript on page 2:

‘‘We first examine the interdependent physicochemical parameters of the lysosomal lumen, including acidity, redox balance, thiol pools, enzymatic content, and metal ions, and explain their hierarchy. The acidic pH (~ 4.5–5.0) provides the foundation by dictating enzyme activity profiles, protonation states, and metal ion solubility. Within this environment, the redox potential, maintained by the glutathione (GSH) GSH/GSSG couple and enzymes such as γ-interferon–inducible lysosomal thiol reductase (GILT), controls thiol reactivity and reactive oxygen species chemistry, which together regulate oxidative and reductive processes.

 Metal ions such as iron (Fe) and copper (Cu) connect these parameters to catalytic transformations and chemical speciation, while the repertoire of acid hydrolases exe-cutes pH-dependent reactions whose rates are further influenced by the local redox and metal status [24]. These factors act as a coordinated network rather than independent features, creating a chemically distinctive microreactor that shapes drug fate.’’

Q3. Line 145-149 Please add further details, i.e. one-two brief sentences reporting a few examples of drugs activated and/or inactivated by acidic pH.

Revision: Thank you for this helpful suggestion. We agree that including specific examples of drugs activated or inactivated by acidic pH would strengthen this section. We have now added a few representative examples illustrating how lysosomal acidity governs drug activation and stability.

The following text has been added to the manuscript (page 4):

‘‘For activation, doxorubicin-bearing hydrazone linkers provide a representative case, as they hydrolyze more rapidly at lysosomal pH to liberate the active drug. For inactivation, weakly basic agents such as sunitinib, doxorubicin, and mitoxantrone become protonated and sequestered within lysosomes, reducing their cytosolic or nuclear availability (Figures 2–3).’’

Q4. Line 182 Could authors provide a reference for the sentence “These ROS oxidize lipids, proteins, and nucleic acids in proximity to the lysosomal membrane.”, or it is just a speculation?

Revision: Thank you for this observation. We agree that the statement requires a supporting reference. We have now added citations to experimental studies demonstrating that lysosome-derived reactive oxygen species can oxidize lipids, proteins, and nucleic acids in the immediate vicinity of the lysosomal membrane.

Revised text and references added to the manuscript (page 6):

These ROS oxidize lipids, proteins, and nucleic acids in proximity to the lysosomal membrane [79–81].

 Q5. Line 185 The sentence “Recent studies have shown that the ferroptosis inducer RAS-selective lethal 3 (RSL3; Figure 5) promotes lipid oxidation specifically within lysosomes, whereas lipophilic antioxidants that localize to lysosomes can suppress ferroptosis by inactivating intralysosomal iron [82, 84].” is unclear and should be re-written.

Revision: Thank you for this helpful comment. We agree that the original sentence required clarification. The revised version now more clearly explains how RSL3 induces lipid peroxidation in lysosomes and how lysosome-targeted antioxidants mitigate this effect by stabilizing intralysosomal iron.

Revised text added to the manuscript (page 7):

Recent studies have shown that the ferroptosis inducer RAS-selective lethal 3 (RSL3; Figure 5) promotes lipid peroxidation within lysosomes by driving iron-dependent oxidative reactions. In contrast, lipophilic antioxidants that accumulate in lysosomes can suppress ferroptosis by neutralizing reactive oxygen species and preventing intralysosomal iron activation [82, 84].

Q6. Line 321 Authors wrote that “These iron-rich lysosomes can act as potential “time bombs” for Fenton chemistry [120]”. However, it is unclear how this phenomenon ca help cancer cell survival and spreading. Coud authors clarify it?

Revision: Thank you for this insightful comment. We agree that the biological implications of iron-rich lysosomes as “time bombs” for Fenton chemistry required further clarification. We have revised the text to explain how this phenomenon contributes to both cancer cell survival and tumor progression.

Revised text added to the manuscript (pages 8-9):

‘‘These iron-rich lysosomes can act as potential “time bombs” for Fenton chemistry [101]. Under physiological conditions, controlled iron redox cycling within lysosomes supports metabolic and signaling processes essential for tumor growth. However, during oxidative stress or partial lysosomal membrane permeabilization, Fen-ton-derived radicals (•OH) can escape into the cytosol, amplifying oxidative signaling, DNA damage, and inflammation. Cancer cells exploit this redox plasticity to tolerate oxidative fluctuations and, paradoxically, to promote survival, proliferation, and met-astatic behavior. The association between lysosomal iron and cell vulnerability is exemplified in ferroptosis, where iron-dependent oxidative reactions trigger lipid peroxidation and cell death [82, 102].’’

Q7. In section 3, the subsection 3.4 appears the logical consequence of subsection 3.2. Subsection 3.3. should be moved at the end of section 3.

Revision: Thank you for this helpful structural suggestion. We agree that subsection 3.4 logically follows subsection 3.2, and that the current order could be improved for better conceptual flow. Accordingly, subsection 3.4 “Enzymatic Hydrolysis: Acid Hydrolases as Catalysts” has been moved to the end of Section 3, after the discussion of 3.3 “Metal Ion Sequestration and Catalysis.”

This reordering ensures a more coherent progression from physicochemical parameters (3.1–3.2) to redox and metal reactivity (3.3), followed by the enzymatic contribution (3.4) as the concluding catalytic aspect of the lysosomal microenvironment. Now this section becomes Section 2.

Q8. In section 5.1, the example of chloroquine is redundant. The behaviour of chloroquine as a weak base was introduced in the previous sessions, with examples of its possible use in combination with other drugs. The session should remind to previous sections and illustrate additional original content on chloroquine, if any.

Revision: Thank you for this helpful observation. We agree that the example of chloroquine in Section 5.1 (now Section 4.1) partially repeated information presented earlier regarding its weak-base trapping behavior. To avoid redundancy, we have revised this subsection to briefly refer to the earlier discussion and to emphasize chloroquine’s broader mechanistic and therapeutic relevance, including its roles in modulating lysosomal function, autophagy, and redox balance.

Revised text added to the manuscript (Section 5.1 (now Section 4.1); page 17):

‘‘As previously discussed in Section 3.1, chloroquine and its derivatives exemplify lysosomotropic agents. Beyond their weak-base trapping properties, these compounds have additional mechanistic relevance. By raising intralysosomal pH, chloroquine interferes with enzyme activation and autophagosome–lysosome fusion, thereby sup-pressing autophagy and sensitizing tumors to chemotherapy and radiotherapy [176, 196]. Moreover, chloroquine-induced lysosomal alkalinization can modulate met-al-dependent redox reactions and disrupt signaling pathways linked to cancer survival. Hydroxychloroquine and other analogues are currently being optimized to improve lysosomal selectivity and minimize systemic toxicity, demonstrating how pharmaco-logical modulation of lysosomal function can complement conventional anticancer strategies.

 Lysosomal targeting can also be achieved through chemical modification. Introducing weak-base functional groups such as morpholine directs otherwise neutral drugs to lysosomes, as morpholine preferentially accumulates in acidic environments. This approach has been used in designing fluorescent probes and could similarly guide therapeutic delivery. Such strategies may be particularly relevant for drugs acting on lysosome-associated pathways, including mammalian target of rapamycin complex 1 (mTORC1), which resides on the lysosomal membrane [199-201]. Targeting inhibitors to this compartment could, in principle, improve selectivity for cancer cells with hyperactive mTOR signaling.

 Lysosomal pH gradients can also be utilized for tumor imaging. Certain radiotracers, including analogs of verdazyl dyes or amine-based probes labeled with car-bon-11 or fluorine-18, accumulate in tumor lysosomes and generate positron emission tomography (PET) signals that correlate with lysosomal acidity or abundance [202]. By fine-tuning a compound’s pKa and lipophilicity, chemists can transform cancer cell lysosomes into drug-concentrating compartments [203]. If the accumulated agent is inherently toxic or membrane-disruptive, its preferential buildup can selectively destroy tumor cells. Achieving selectivity over normal tissues remains a challenge, but differences in perfusion, pH, and drug retention between tumor and normal tissue may provide exploitable therapeutic windows.’’

Q9. In Table 2, what do authors mean with the definition “Theranostics” as Strategy? It is unclear.

Revision: Thank you for this helpful comment. We agree that the term “Theranostics” in Table 2 required clarification. To ensure clarity, we have replaced the term “Theranostics” with “Therapeutic + Diagnostic (Theranostic) Approach” in the Strategy column of Table 2.

Additionally, the table caption has been expanded to include a brief explanatory note, as follows:

‘‘Table 2. Drug Design and Therapeutic Strategies Exploiting Lysosomal Chemistry. This table summarizes representative approaches that utilize lysosomal properties for therapeutic advantage. The term Therapeutic + Diagnostic (Theranostic) Approach refers to dual-function agents that combine therapeutic and diagnostic capabilities within a single molecular or nanostructured system, enabling simultaneous drug delivery, imaging, and treatment monitoring.’’

Q10. Sections 6.2 to 6.4: The organization in such small subsections can be avoided, as it increase the overall length without improving the clarity.

Revision: Thank you for this constructive suggestion. We agree that the separation of Sections 6.2–6.4 (now Section 5.2) into multiple small subsections was unnecessary and increased length without improving clarity. Accordingly, we have merged these subsections into a single integrated section (now revised as Section 5.2: “Peptide- and Polymer–Drug Conjugates: Enzyme-Cleavable Systems and Design Considerations”).

This consolidated section now discusses peptide- and polymer-based drug conjugates together, followed by the shared design challenges, ensuring a more coherent and compact presentation.

Revised integrated section for the manuscript (page 19):

‘‘5.2. Peptide- and Polymer–Drug Conjugates: Enzyme-Cleavable Systems and Design Con-siderations

Smaller analogs of antibody–drug conjugates (ADCs), known as peptide–drug conjugates (PDCs), employ tumor-homing peptides instead of antibodies to achieve targeted delivery [220]. Following receptor-mediated endocytosis, lysosomal proteases cleave the linker to release the active payload. For example, the cytolytic peptide melittin has been linked to a matrix metalloproteinase (MMP)-sensitive sequence to enable activation specifically in MMP-rich tumor environments [221]. Although MMPs often act extracellularly, similar strategies can be applied to lysosomal proteases such as cathepsin D, which functions optimally in the acidic lysosomal lumen and can trig-ger intracellular activation.

 Synthetic polymer–drug conjugates also employ enzyme-responsive linkers for selective release within lysosomes [222]. Notably, N-(2-hydroxypropyl)-methacrylamide (HPMA) copolymers use Gly–Phe–Leu–Gly spacers that are cleaved by cathepsin B [223]. These systems mirror ADC linker design principles but replace the antibody with a synthetic polymer scaffold, providing great-er versatility in molecular weight, composition, and pharmacokinetic control.

 Enzyme-cleavable conjugates rely critically on the activity of specific lysosomal enzymes, which can vary among tumor types and microenvironments due to differences in pH, oxygen tension, and metabolic stress [150]. While cathepsins are generally active under acidic conditions, tumor hypoxia or lysosomal rupture may affect their localization or catalytic efficiency. Off-target activation in normal tissues remains a potential limitation; however, ADCs and PDCs maintain selectivity primarily through receptor- or antigen-mediated endocytosis, minimizing systemic exposure. Collectively, these enzyme-cleavable systems exemplify how lysosomal protease activity can be harnessed for precise intracellular drug activation.’’

 Q11. The organization of section 7 is awkward: the title is “Emerging Concepts: Lysosome-Targeting Chimeras”, but the subsection 7.1 title is “Overcoming Lysosomal Drug Sequestration: P-gp Repurposing and Lysosome-Targeted 728 Combinations”.

Revision: Thank you for this valuable suggestion. We agree that the organization of Section 7 (now Section 6) required clarification. The subsection title “Overcoming Lysosomal Drug Sequestration: P-gp Repurposing and Lysosome-Targeted Combinations” did not align smoothly with the main heading “Emerging Concepts: Lysosome-Targeting Chimeras.”

To improve coherence and logical flow, we have revised the structure as follows:

The main section title has been broadened to “Emerging Concepts in Lysosomal Modulation and Targeted Chimeras.”

Subsection 7.1 (now Section 6.1) has been renamed to “Overcoming Lysosomal Drug Sequestration and Combination Strategies.”

This new organization clearly separates (i) emerging therapeutic approaches that modulate lysosomal behavior, from (ii) novel molecular designs such as LYTACs that directly harness lysosomal degradation pathways.

The following section has been updated in the revised manuscript, Section 6, page 20:

 ‘‘6. Emerging Concepts in Lysosomal Modulation and Targeted Chimeras

 An exciting frontier in drug design involves redirecting or exploiting lysosomal machinery for therapeutic benefit. Among these strategies, lysosome-targeting chime-ras (LYTACs) have emerged as bifunctional molecules capable of recruiting extracellular or membrane proteins to lysosomes for degradation through receptors such as the asialoglycoprotein receptor [224-227]. Conceptually, LYTACs are the lysosomal counterparts of proteolysis-targeting chimeras (PROTACs) that drive proteasomal degradation [228]. Although still early in development, LYTACs represent a powerful way to harness the lysosomal degradation pathway for eliminating oncogenic or pathogenic membrane proteins.

 Harnessing lysosomal enzymes for controlled drug release is now validated by several approved therapeutics and many experimental candidates. This approach leverages the abundance and catalytic efficiency of lysosomal hydrolases, which remain sequestered from the extracellular space, thereby providing intrinsic selectivity. Future work will likely refine enzyme-cleavable linkers to improve tissue specificity, activation kinetics, and stability, further consolidating lysosomal enzymology as a foundation for precision drug delivery.

 6.1. Overcoming Lysosomal Drug Sequestration and Combination Strategies

An emerging theme in cancer pharmacology is to exploit the very mechanisms that confer drug resistance. Lysosomal sequestration, once viewed purely as a liability, is now being reimagined as a therapeutic opportunity. Multiple strategies are being developed to counter or repurpose this process, including combination therapies that release trapped agents and P-gp-targeted approaches that redirect drug transport to sensitize cancer cells.’’

 Q12. Section 7.1.5 begins with “Cancer cells with elevated lysosomal iron are particularly susceptible to ferroptosis, a form of cell death driven by iron-dependent lipid peroxidation. This is redundant, as it has been already explained in the text.

Revision: Thank you for this helpful comment. We agree that the opening sentence of Section 7.1.5 (now Section 6.1.5) was redundant. We have removed the definitional statement and revised the subsection to begin directly with the therapeutic context.

The following section has been updated in the revised manuscript, Section 6.1.5, page 23:

‘‘6.1.5. Sensitizing Cells to Ferroptosis via Lysosomal Iron

 CD44-high, mesenchymal-like tumor cells harbor large lysosomal iron stores that can be therapeutically exploited [233]. Strategies include delivering iron-binding compounds that liberate reactive iron within lysosomes or employing iron-oxide nanoparticles that dissolve under acidic conditions to fuel localized ROS generation. These interventions shift the lysosomal milieu toward a pro-ferroptotic state, amplifying oxidative damage and triggering selective death of iron-rich cancer cells.

 Collectively, these strategies represent a paradigm shift in cancer drug design: lysosomal sequestration is no longer viewed merely as a passive sink that diminishes efficacy but as an active, targetable process. By leveraging lysosomal accumulation, redox chemistry, and membrane dynamics, researchers are transforming a classical resistance mechanism into a therapeutic frontier.’’

Q13. The section “Future perspectives” is too long and fragmented: it should more concise and focused.

Revision: Thank you for this helpful suggestion. We agree that the Future Perspectives section was lengthy and fragmented. We have now condensed and unified it into a shorter, more focused version that emphasizes tumor selectivity, resistance modulation, cell-death strategies, and translational prospects, improving clarity and readability.

The following section has been revised in the manuscript (pages 24–25):

‘‘7. Future Perspectives

 Viewing the lysosome as a chemical reactor opens new possibilities for cancer therapy. Future research will focus on improving tumor selectivity, overcoming drug resistance, and translating lysosomal targeting into safe and effective treatments.

 Selective targeting will depend on identifying features that distinguish cancer lysosomes from those in normal cells, such as altered acidity, membrane proteins, or metal content. Designing ligands, peptides, or nanocarriers that recognize these traits could achieve precise drug delivery to tumor lysosomes.

 Because lysosomal sequestration contributes to resistance, combination strategies that adjust lysosomal pH or disrupt its membrane can restore sensitivity to therapy. Agents such as DpC and doxorubicin already illustrate this potential. Similarly, activating lysosome-dependent death pathways such as ferroptosis or controlled mem-brane rupture may provide alternatives for apoptosis-resistant tumors. Advances in imaging and molecular profiling will allow mapping of lysosomal activity across cancers, supporting more personalized drug design. Smart carriers that respond to pH, redox state, or enzyme activity will enable precise intracellular release. Biomarkers of lysosomal function, including plasma cathepsins or imaging tracers, can help monitor safety.

 Progress in this field will rely on collaboration between chemists, biologists, pharmacologists, and clinicians. Together, these efforts will transform the lysosome from a degradative organelle into a controllable site for selective drug activation and mechanism-based cancer therapy.’’

Reviewer 2 Report

Comments and Suggestions for Authors

I want to thank the Editors for the opportunity to review this insightful and well-written manuscript, which explores the emerging view of the lysosome as a chemically dynamic organelle that influences the behavior of anticancer drugs. The topic is timely and highly relevant, integrating aspects of cell biology, pharmacology, and chemical biology to reframe the lysosome as a microenvironmental reactor rather than a passive degradative compartment. The manuscript is generally well organized and clearly written. Overall, I find the review to be of substantial scientific merit and suitable for publication after the authors address a few comments aimed at improving clarity, completeness, and conceptual depth.

GENERAL COMMENTS

  1. A graphical abstract is necessary for this manuscript, given the length and complexity of the content. A concise visual summary would help readers quickly grasp the lysosome's role as a chemical reactor and its multifaceted influence on the behavior of anticancer drugs. I recommend that the authors design a graphical abstract based on the order of their manuscript’s sections. I recommend using BioRender to design this graphical abstract with the utmost professionalism.
  2. The quality of the figures should be improved. The current use of colorful backgrounds and low-resolution vector elements detracts from the manuscript's professional appearance. Given the depth and quality of the review, the figures should be redesigned with a cleaner aesthetic, a consistent color palette, and higher vector resolution to match the scholarly standards of the text. I recommend using BioRender to achieve these goals effectively and with greater quality.
  3. I believe that the title of this manuscript should better convey its purpose. In the introduction, it is disclosed that the current review aims to address ‘how a deeper chemical understanding of lysosomes could guide the development of next-generation, more selective and effective anticancer therapies.’ The current title does not reflect the purpose of this manuscript and, therefore, must be rewritten.

METHODS

  1. A brief Methods section is recommended, even for a review article, as a substantial amount of information, data interpretation, and literature synthesis is presented. This section should outline the search strategy, the databases used, the inclusion and exclusion criteria, and the timeframe of the literature considered, to ensure transparency and reproducibility of the review process. If you cannot add “Methods” to the main manuscript, at least include it in the appendix.

DISCUSSION

  1. The authors mention that compounds such as chloroquine, hydroxychloroquine, DpC, COTI-2, and triapine are currently being evaluated in clinical trials. To strengthen this section and provide a more straightforward overview of the translational relevance, the manuscript should include summary tables of recent or ongoing trials, organized in PICO-style columns (Population, Intervention, Comparator, Outcomes), along with trial phase, cancer type, status, and references. This addition would improve the readability and accessibility of key clinical information, highlight the novelty and therapeutic potential of these agents within the lysosomal-targeting framework, and facilitate future updates or meta-analyses by providing a structured overview of the current clinical landscape.
  2. The manuscript would benefit from better integration of the mechanistic aspects of lysosomal chemistry. Currently, pH, redox balance, metal ion chemistry, and enzyme activity are mainly discussed in isolation. The authors should aim to present a more unified conceptual model or schematic showing how these features converge to influence drug sequestration, activation, and efficacy.
  3. It is recommended that the authors include a more in-depth and illustrative discussion on lysosomal heterogeneity between normal and cancer cells. Differences in pH, enzyme content, metal ion concentrations, and other lysosomal features can strongly influence drug behavior and therapeutic outcomes.

Overall, this manuscript represents a significant contribution to the field by reframing the lysosome as a dynamic chemical reactor with essential implications for anticancer therapy. After addressing the points listed above —particularly figure quality, title, mechanistic integration, and clinical trial representation —the review will be clearer, more comprehensive, and more accessible to readers. I recommend publication pending these revisions, as the work will provide valuable insights into lysosomal biology, drug development, and therapeutic resistance.

I also want to recommend this manuscript for highlighting on the journal’s website. I believe this review will attract many citations over the long term.

I would also like to commend the authors for their work in ensuring that only 7% similarity index was found in their manuscript. As a review article with a substantial amount of evidence gathered, it would be expected that the similarity index would be much higher. You did a great job of ensuring that your manuscript is as original as possible.

I appreciate your consideration. I hope you have a great revision process.

With gratitude,

Author Response

14th November 2025

Referee 2: Recommendation: Minor revisions and stated: “This insightful and well-written manuscript, which explores the emerging view of the lysosome as a chemically dynamic organelle that influences the behavior of anticancer drugs. The topic is timely and highly relevant, integrating aspects of cell biology, pharmacology, and chemical biology to reframe the lysosome as a microenvironmental reactor rather than a passive degradative compartment. The manuscript is generally well organized and clearly written. Overall, I find the review to be of substantial scientific merit and suitable for publication after the authors address a few comments aimed at improving clarity, completeness, and conceptual depth.”

We thank the reviewers for their very constructive and insightful feedback. We appreciate the opportunity to revise our manuscript and believe that the revisions, along with the addition of the new section, and table have substantially enhanced its clarity and impact.

We are also grateful to you and the reviewers for your thoughtful evaluation. We are confident that the revised version satisfactorily addresses all concerns raised and will be of significant interest to the readers of International Journal of Molecular Sciences.

Kind regards,

Dr. Mahendiran Dharmasivam, B.Sc., M.Sc., B.Ed., Ph.D., MRACI

National Breast Cancer Foundation (NBCF) Fellow

Institute for Biomedicine and Glycomics, Griffith University Gold Coast

Southport QLD 4215, Australia

Phone: +61-7555-27502; Email: m.dharmasivam@griffith.edu.au

Referee 2: Recommendation: Minor Revision

 Q1. A graphical abstract is necessary for this manuscript, given the length and complexity of the content. A concise visual summary would help readers quickly grasp the lysosome's role as a chemical reactor and its multifaceted influence on the behavior of anticancer drugs. I recommend that the authors design a graphical abstract based on the order of their manuscript’s sections. I recommend using BioRender to design this graphical abstract with the utmost professionalism.

 Revision: We agree that a concise visual summary will greatly aid readers. A professional graphical abstract has been included to highlight the lysosome’s central role as a chemical reactor and summarize key mechanistic and therapeutic concepts discussed throughout the manuscript. The illustration depicts:

  • pH-dependent weak-base trapping of drugs,
  • metal-mediated Fenton/Fenton-like processes generating ROS,
  • pH-triggered release of antibody–drug conjugates (ADCs), and
  • the resulting cell death pathways.

This visual encapsulates the main mechanistic themes in a clear and accessible way, helping readers quickly appreciate the lysosome’s chemical and therapeutic relevance.

Q2. The quality of the figures should be improved. The current use of colorful backgrounds and low-resolution vector elements detracts from the manuscript's professional appearance. Given the depth and quality of the review, the figures should be redesigned with a cleaner aesthetic, a consistent color palette, and higher vector resolution to match the scholarly standards of the text. I recommend using BioRender to achieve these goals effectively and with greater quality.

 Revision: We thank the reviewer for this valuable comment. All figures have been redesigned using BioRender to ensure a consistent, high-quality, and professional visual presentation. Each figure now features:

  • a clean, neutral background to improve clarity and focus;
  • a harmonized color palette reflecting mechanistic continuity across sections; and
  • high-resolution vector graphics suitable for both digital and print display.

These revisions enhance readability and visual consistency while aligning the figure aesthetics with the scholarly tone and depth of the manuscript.

Q3. I believe that the title of this manuscript should better convey its purpose. In the introduction, it is disclosed that the current review aims to address ‘how a deeper chemical understanding of lysosomes could guide the development of next-generation, more selective and effective anticancer therapies.’ The current title does not reflect the purpose of this manuscript and, therefore, must be rewritten.

 Revision: We appreciate the reviewer’s perspective. We respectfully retain the current title because it captures the paper’s central thesis, the lysosome as a chemical reactor, which serves as the unifying conceptual framework linking mechanisms (weak-base trapping, metal-mediated chemistry, enzyme processing) and therapeutic modalities (small molecules, metal complexes, and ADCs). A broader conceptual title ensures accessibility to diverse research audiences and avoids over-specifying any single mechanism or drug class.

To address the reviewer’s concern regarding clarity of purpose, we have strengthened the framing throughout the manuscript:

  • Abstract (last sentence, revised): now states that we translate chemical principles of the lysosome into design rules for next-generation, more selective anticancer strategies. 
  • Keywords (updated): now emphasize lysosomotropic design, drug–lysosome interactions, metal-mediated ROS, and therapeutic design principles, improving discoverability for readers seeking translational insights.

These revisions make the manuscript’s intent more explicit while retaining a concise, concept-driven title that reflects its integrative chemical and therapeutic scope.

Q4. A brief Methods section is recommended, even for a review article, as a substantial amount of information, data interpretation, and literature synthesis is presented. This section should outline the search strategy, the databases used, the inclusion and exclusion criteria, and the timeframe of the literature considered, to ensure transparency and reproducibility of the review process. If you cannot add “Methods” to the main manuscript, at least include it in the appendix.

 Revision: We thank the reviewer for this thoughtful suggestion. Because this article is a conceptual and mechanistic review rather than a systematic analysis, we have not added a formal Methods section. However, we have ensured transparency by clarifying in the Introduction that the discussion integrates both recent and seminal studies selected for their relevance to lysosomal chemistry, redox biology, and therapeutic design.

This concise statement defines the literature scope and selection rationale while maintaining the narrative structure and scholarly flow appropriate for a topical review in International Journal of Molecular Sciences.

Q5. The authors mention that compounds such as chloroquine, hydroxychloroquine, DpC, COTI-2, and triapine are currently being evaluated in clinical trials. To strengthen this section and provide a more straightforward overview of the translational relevance, the manuscript should include summary tables of recent or ongoing trials, organized in PICO-style columns (Population, Intervention, Comparator, Outcomes), along with trial phase, cancer type, status, and references. This addition would improve the readability and accessibility of key clinical information, highlight the novelty and therapeutic potential of these agents within the lysosomal-targeting framework, and facilitate future updates or meta-analyses by providing a structured overview of the current clinical landscape.

 Revision: We thank the reviewer for this excellent suggestion. We agree that summarizing the translational relevance of these lysosome-targeting agents will enhance clarity and accessibility. Accordingly, we have added a concise summary table in the revised manuscript that lists key recent or ongoing clinical trials for representative compounds, including chloroquine, hydroxychloroquine, DpC, COTI-2, and triapine.

To maintain readability and journal formatting standards, the table (table 3; page 22) includes the following information:

  • Cancer type and population,
  • Intervention description,
  • Trial phase and current status, and
  • Clinical identifier and reference.

This streamlined table provides a clear overview of the clinical translation landscape while preserving focus on the chemical principles and lysosomal mechanisms discussed throughout the review.

Q6. The manuscript would benefit from better integration of the mechanistic aspects of lysosomal chemistry. Currently, pH, redox balance, metal ion chemistry, and enzyme activity are mainly discussed in isolation. The authors should aim to present a more unified conceptual model or schematic showing how these features converge to influence drug sequestration, activation, and efficacy.

 Revision: Thank you for this excellent suggestion. We agree that the mechanistic aspects of lysosomal chemistry (pH, redox balance, metal ions, and enzyme activity) were previously presented in a compartmentalised manner. In response, we have now added a unifying integrative paragraph at the end of the introductory section of Section 3 (now Section 2), immediately before subsection 3.1 (now Section 2.1) (Acidic pH and Protonation Dynamics). This paragraph explicitly links acidity, redox cycling, metal chemistry, and enzymatic or linker-mediated cleavage into a single conceptual framework, clarifying how these features converge to determine drug sequestration, activation, and lysosomal-dependent cytotoxicity.

We have also updated the caption of Figure 1 so that it visually reflects this integrated model.

The following statement has been added to the revised manuscript on pages 3-4:

‘‘Overall, the lysosome functions as an integrated chemical reactor rather than a collection of isolated conditions. Acidic pH generated by the V-ATPase sets the proto-nation state of weak bases, controls hydrolase activity, and influences the solubility and speciation of Fe and Cu (Figure 1). Within this acidic lumen, redox-active metals, low-molecular-weight thiols, and hydrogen peroxide interact to sustain Fenton and Fenton-like reactions that generate ROS. At the same time, pH-sensitive linkers and enzyme substrates in antibody–drug conjugates and other carriers are cleaved during endosomal–lysosomal trafficking, releasing active payloads. As summarized in Figure 1, these interconnected processes determine whether drugs are sequestered and functionally inactivated, locally activated within lysosomes, or cooperate with ROS to damage membranes and trigger cell death.’’

Q7. It is recommended that the authors include a more in-depth and illustrative discussion on lysosomal heterogeneity between normal and cancer cells. Differences in pH, enzyme content, metal ion concentrations, and other lysosomal features can strongly influence drug behavior and therapeutic outcomes.

 Revision: Thank you for this valuable recommendation. We agree that lysosomal heterogeneity between normal and cancer cells is an important determinant of drug disposition and therapeutic outcome. To address this, we have now added a dedicated paragraph in Section 3 (now Section 2) that provides a clearer and more detailed comparison of lysosomal properties in normal versus malignant cells. This new text discusses differences in luminal pH, V-ATPase activity, lysosomal volume, hydrolase expression, and metal ion content, and explains how these features create a more reactive and drug-responsive lysosomal environment in cancer cells. These additions clarify why tumour lysosomes show enhanced weak-base trapping, greater redox activity, and heightened susceptibility to lysosome-dependent cytotoxic mechanisms.

The following paragraph has been added to Section 2 in the revised manuscript (page 4):

‘‘Importantly, lysosomal properties differ substantially between normal and cancer cells, and these variations strongly influence drug behaviour. Many tumours display deeper luminal acidification, increased V-ATPase activity, enlarged lysosomal volume, and elevated expression of cathepsins and other hydrolases, which together enhance weak-base trapping and accelerate degradation or activation of pH- or enzyme-sensitive linkers. Cancer cells also accumulate larger pools of labile iron and copper due to heightened autophagy and altered metal metabolism, predisposing their lysosomes to Fenton-type ROS production and redox cycling. In contrast, normal cells maintain tighter control over luminal pH, metal availability, and hydrolase expression, creating a less reactive environment. These tumour-specific features establish a chemically more potent and drug-responsive lysosomal compartment that shapes differential sequestration, redox sensitivity, and susceptibility to lysosome-dependent cell death.’’

Q9. I also want to recommend this manuscript for highlighting on the journal’s website. I believe this review will attract many citations over the long term.

 Revision: We are deeply grateful for the reviewer’s generous endorsement and confidence in the long-term impact of our work. It is very encouraging to know that the manuscript’s conceptual depth and translational relevance are recognized. We sincerely appreciate this positive recommendation and the reviewer’s support for highlighting the article on the International Journal of Molecular Sciences website.

Q10. I would also like to commend the authors for their work in ensuring that only 7% similarity index was found in their manuscript. As a review article with a substantial amount of evidence gathered, it would be expected that the similarity index would be much higher. You did a great job of ensuring that your manuscript is as original as possible.

 Revision: We sincerely thank the reviewer for this kind and encouraging comment. We made a concerted effort to ensure that the manuscript reflects original synthesis and interpretation of the literature rather than direct textual overlap. We greatly appreciate the reviewer’s recognition of this aspect, which reinforces our commitment to maintaining the highest standards of scientific integrity and originality in review writing.

Q11. I appreciate your consideration. I hope you have a great revision process.

 Revision: We sincerely thank the reviewer for their kind words and positive encouragement. We truly appreciate the constructive feedback provided throughout the review, which has helped us strengthen the clarity, depth, and overall quality of the manuscript.

Round 2

Reviewer 1 Report

Comments and Suggestions for Authors

The manuscript has addressed most issues raised.

A few points

In the sentence “lipophilic antioxidants that accumulate in lysosomes can suppress ferroptosis”, please list one or two of these “lipophilic antioxidants”

Pages 8-9 The sentence “The association between lysosomal iron and cell vulnerability is exemplified in ferroptosis, where iron-dependent oxidative reactions trigger lipid peroxidation and cell death” should be moved after “Fen-ton-derived radicals (•OH) can escape into the cytosol, amplifying oxidative signaling, DNA damage, and inflammation”. Indeed, putting the sentence after the explanation of “how these damages can paradoxically train cancer cell to survive” is confusing and the concept that these damages can paradoxically train cancer cell to survive still remain adding experimental support with suitable references  

Author Response

22nd November 2025

Referee 1: Recommendation: Minor revisions and stated: “''addressed most of the issues raised.”

We again thank the reviewer for their very constructive and insightful feedback. We appreciate the opportunity to revise our manuscript and believe that the revisions, along with the addition of the new section and table, have substantially enhanced its clarity and impact.

We are also grateful to you and the reviewer for your thoughtful evaluation. We are confident that the revised version satisfactorily addresses all concerns raised and will be of significant interest to the readers of the International Journal of Molecular Sciences.

Kind regards,

Dr. Mahendiran Dharmasivam, B.Sc., M.Sc., B.Ed., Ph.D., MRACI

National Breast Cancer Foundation (NBCF) Fellow

Institute for Biomedicine and Glycomics, Griffith University Gold Coast

Southport QLD 4215, Australia

Phone: +61-7555-27502; Email: m.dharmasivam@griffith.edu.au

Referee 1: Recommendation: Minor Revision

Q1. In the sentence “lipophilic antioxidants that accumulate in lysosomes can suppress ferroptosis”, please list one or two of these “lipophilic antioxidants.

Revision: We thank the reviewer for this helpful suggestion. Lipophilic, membrane-embedded antioxidants that suppress ferroptosis include Coenzyme Q10 (CoQ10) and vitamin E (α-tocopherol). Both function as radical-trapping antioxidants and prevent iron-dependent lipid peroxidation. Synthetic RTAs such as ferrostatin-1 and liproxstatin-1 also act through this mechanism. We have now added representative examples to the revised text.

The following revised paragraph has been added to the manuscript on page 7:

In contrast, lipophilic antioxidants such as Coenzyme Q10 (CoQ10) [84, 85] or vitamin E (α-tocopherol) [86, 87], which act as membrane-embedded radical-trapping antioxi-dants, can suppress ferroptosis by neutralizing reactive oxygen species and preventing intralysosomal iron activation [82, 88]. Synthetic radical-trapping antioxidants such as ferrostatin-1 and liproxstatin-1 also inhibit ferroptosis by blocking lipid peroxidation. Similarly, a synthetic molecule named fentomycin-1 was designed to activate lysoso-mal iron directly, triggering extensive phospholipid oxidation and selective death of iron-rich cancer cells [82]. These findings highlight that the site of redox activity is crucial: the lysosome is not merely a bystander in oxidative stress but can serve as a deliberate target for pro-oxidant cancer therapies.

Q2. Pages 8-9 The sentence “The association between lysosomal iron and cell vulnerability is exemplified in ferroptosis, where iron-dependent oxidative reactions trigger lipid peroxidation and cell death” should be moved after “Fen-ton-derived radicals (•OH) can escape into the cytosol, amplifying oxidative signaling, DNA damage, and inflammation”. Indeed, putting the sentence after the explanation of “how these damages can paradoxically train cancer cell to survive” is confusing and the concept that these damages can paradoxically train cancer cell to survive still remain adding experimental support with suitable references.

Revision: We thank the reviewer for this insightful comment. We agree that the sentence describing ferroptosis should appear immediately after the explanation of Fenton-derived radical escape, and we have relocated this sentence accordingly to improve the logical flow.

As suggested, we also clarified how sub-lethal lysosomal iron–driven oxidative stress can paradoxically promote cancer cell survival. We incorporated experimental support with suitable references demonstrating that chronic, low-level oxidative stress activates NRF2-dependent antioxidant programmes and enhances tumour aggressiveness.

The following revised paragraph has been added to the manuscript on page 9:

‘‘Under physiological conditions, controlled iron redox cycling within lysosomes sup-ports metabolic and signalling processes essential for tumour growth. However, during oxidative stress or partial lysosomal membrane permeabilization, Fenton-derived radicals (•OH) can escape into the cytosol, amplifying oxidative signalling, DNA damage, and inflammation. The association between lysosomal iron and cell vulnerability is exemplified in ferroptosis, where iron-dependent oxidative reactions trigger lipid peroxidation and cell death [82, 106]. Importantly, chronic exposure to sub-lethal oxidative stress can paradoxically promote cancer cell adaptation by activating NRF2-driven antioxidant programmes, enhancing metabolic flexibility, and supporting tumour proliferation, invasion, and therapy resistance [107, 108].’’